# DiffComplete: Diffusion-based Generative 3D Shape Completion

**Ruihang Chu**[1]     **Enze Xie** [2]     **Shentong Mo**[3]
**Zhenguo Li**[2]    **Matthias Nießner**[4]    **Chi-Wing Fu**[1]    **Jiaya Jia**[1,5]

[1]The Chinese University of Hong Kong    [2]Huawei Noah's Ark Lab
[3]MBZUAI    [4]Technical University of Munich    [5]SmartMore
https://ruihangchu.com/diffcomplete.html

## Abstract

We introduce a new diffusion-based approach for shape completion on 3D range scans. Compared with prior deterministic and probabilistic methods, we strike a balance between realism, multi-modality, and high fidelity. We propose DiffComplete by casting shape completion as a generative task conditioned on the incomplete shape. Our key designs are two-fold. First, we devise a hierarchical feature aggregation mechanism to inject conditional features in a spatially-consistent manner. So, we can capture both local details and broader contexts of the conditional inputs to control the shape completion. Second, we propose an occupancy-aware fusion strategy in our model to enable the completion of multiple partial shapes and introduce higher flexibility on the input conditions. DiffComplete sets a new SOTA performance (*e.g.*, 40% decrease on $l_1$ error) on two large-scale 3D shape completion benchmarks. Our completed shapes not only have a realistic outlook compared with the deterministic methods but also exhibit high similarity to the ground truths compared with the probabilistic alternatives. Further, DiffComplete has strong generalizability on objects of entirely unseen classes for both synthetic and real data, eliminating the need for model re-training in various applications.

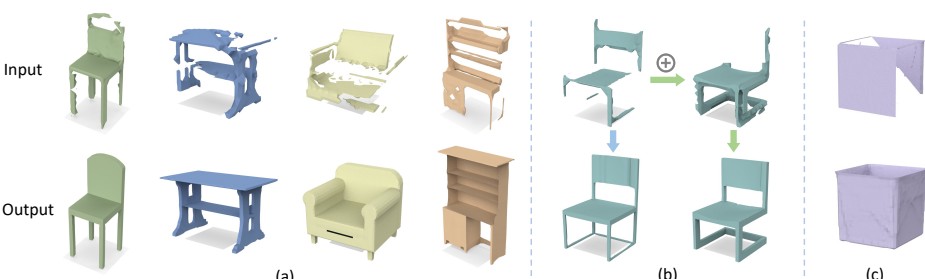

Figure 1: Our method is able to (a) produce realistic completed shapes from partial scans, (b) incorporate multiple incomplete scans (denoted by the plus symbol) to improve the completion accuracy, and (c) directly generalize to work on real objects of unseen classes without finetuning.

## 1 Introduction

The advent of affordable range sensors, such as the Microsoft Kinect and Intel RealSense, has spurred remarkable progress in 3D reconstruction [1], facilitating applications in content creation, mixed reality, robot navigation, and more. Despite the improved reconstruction quality [2–5], typical scanning sessions cannot cover every aspect of the scene objects. Hence, the reconstructed 3D models often have incomplete geometry, thereby limiting their practical usage in downstream applications.

37th Conference on Neural Information Processing Systems (NeurIPS 2023).

To fully unlock the potential of 3D reconstruction for supporting assorted applications, it is essential to address the challenges of completing incomplete shapes.

Effectively, a shape completion should produce shapes that are *realistic*, *probabilistic*, and *high-fidelity*. First, the produced shapes should look plausible without visual artifacts. Second, considering the under-determined nature of completion, it is desirable for the model to generate diverse, multi-modal outputs when filling in the missing regions, to improve the likelihood of obtaining a superior shape and enable creative use. Third, while multiple outputs encourage the model's generative ability, maintaining effective control over the completion results is crucial to ensure coherent reconstructions that closely resemble the ground-truth shapes.

Current approaches to 3D shape completion fall into deterministic and probabilistic paradigms. The former class excels at aligning predictions with ground truths, owing to their determined mapping functions and full supervision. However, this can expose the models to a higher risk of over-fitting, leading to undesirable artifacts, such as rugged surfaces, especially on unseen objects. On the other hand, probabilistic approaches formulate the shape completion as a conditional generation task to produce plausible results, paired with techniques like autoregressive model [6], adversarial training [7–9], or diffusion models [10–13]. These approaches mainly focus on cases that prioritize completion diversity, for instance, filling in large missing regions cropped out by 3D boxes, where the algorithm has more freedom to explore various plausible shapes. However, when completing shapes obtained from range scans/reconstructions, whose geometries are often contaminated with noise, having varying degrees of incompleteness (see row 1 in Fig. 1), a promising goal is to recover the ground-truth scanned objects as accurately as possible. Therefore, relying solely on previous probabilistic approaches may compromise the completion accuracy and leads to low-fidelity results.

In this work, we focus on completing shapes acquired from range scans, aiming to produce *realistic* and *high-fidelity* completed shapes, while considering also the *probabilistic* uncertainty. We approach this task as a conditional generation with the proposed diffusion model named DiffComplete. To address the accuracy challenge typically in probabilistic models, we introduce a key design that incorporates range scan features in a hierarchical and spatially-consistent manner to facilitate effective controls. Additionally, we propose a novel occupancy-aware fusion strategy that allows taking multiple partial shapes as input for more precise and controllable completions.

Specifically, we formulate the diffusion process on a truncated distance field in a volumetric space. This representation allows us to encode both complete and incomplete shapes into multi-resolution structured feature volumes, thereby enabling their interactions at various network levels. At each level, features are aggregated based on voxel-to-voxel correspondence, enabling precise correlation of the difference in partialness between the shapes. By leveraging multi-level aggregation, the completion process can be controlled by both local details and broader contexts of the conditional inputs. It effectively improves the completion accuracy and network generalizability, as illusrated in Sec. 4.5.

To condition the completion on multiple partial shapes, our occupancy-aware fusion strategy adopts the occupancy mask as a weight to adaptively fuse all observed geometry features from various incomplete shapes, and then employ the fused features to control the completion process. For robustness, such an operation is performed in a multi-scale feature space. With our novel design, switching between single and multiple conditions can be efficiently achieved by finetuning an MLP layer. As Fig. 1(b) and 6 show, using multiple conditions for completion progressively refines the final shapes towards the ground truths.

DiffComplete sets a new state-of-the-art performance on two large-scale shape completion benchmarks [14, 15] in terms of completion accuracy and visual quality. Compared to prior deterministic and probabilistic approaches, we not only generate multimodal plausible shapes with minimal artifacts but also present high shape fidelity relative to the ground truths. Further, DiffComplete can directly generalize to work on unseen object categories. Without specific zero-shot designs, it surpasses all existing approaches for both synthetic and real-world data, allowing us to eliminate the need for model re-training in various applications. To sum up, our contributions are listed as follows:

- We introduce a diffusion model to complete 3D shapes on range scans, enabling realistic, probabilistic, and high-fidelity 3D geometry.
- We enhance the completion accuracy with two designs. For a single condition, we propose a hierarchical feature aggregation strategy to control the outputs. For multiple conditions, we introduce an occupancy-aware fusion strategy to incorporate more shape details.

- We show SOTA performance on shape completion on both novel instances and entirely unseen object categories, along with an in-depth analysis on a range of model characteristics.

## 2   Related Work

**RGB-D reconstruction.** Traditional methods rely on geometric approaches for 3D reconstruction [16–20]. A pioneering method [21] proposes a volumetric fusion strategy to integrate multiple range images into a unified 3D model on truncated signed distance fields (TSDF), forming the basis for many modern techniques like KinectFusion [22, 23], VoxelHashing [24], and BundleFusion [2]. Recent learning-based approaches further improve the reconstruction quality with fewer artifacts [25–28, 3], yet the intrinsic occlusions and measurement noise of 3D scans constrain the completeness of 3D reconstructions, making them still less refined than manually-created assets.

**3D shape completion** is a common post-processing step to fill in missing parts in the reconstructed shapes. Classic methods mainly handle small holes and geometry primitives via Laplacian hole filling [29–31] or Poisson surface reconstruction [32, 33]. Another line exploits the structural regularities of 3D shapes, such as the symmetries, to predict the unobserved data [34–38].

The availability of large 3D data has sparked retrieval-based methods [39–42] and learning-based fitting methods [43, 44, 14, 45–49]. The former retrieves the shapes from a database that best match the incomplete inputs, whereas the latter minimizes the difference between the network-predicted shapes and ground truths. 3D-EPN [14], for instance, proposes a 3D encoder-decoder architecture to predict the complete shape from partial volumetric data. Scan2Mesh [50] converts range scans into 3D meshes with a direct optimization on the mesh surface. PatchComplete [15] further leverages local structural priors for completing shapes of unseen categories.

Generative methods, e.g., GANs [51, 9, 52, 7, 8] and AutoEncoders [53], offer an alternative to shape completion. While some generative models can generate diverse global shapes given a partial input, they potentially allow for a high generation freedom and overlook the completion accuracy. IMLE [54], for instance, adopts an Implicit Maximum Likelihood Estimation technique to specially enhance structural variance among the generated shapes. Distinctively, we formulate a diffusion model for shape completion. Our method mainly prioritizes fidelity relative to ground truths while preserving output diversity. Compared to both generative and fitting-based paradigms, our method also effectively reduces surface artifacts, producing more realistic and natural 3D shapes. In addition, we show superior generalization ability on completing objects of novel classes over SOTAs.

**Diffusion models for 3D generation.** Diffusion models [55–62] have shown superior performance in various generation tasks, outperforming GANs' sample quality while preserving the likelihood evaluation property of VAEs. When adopted in 3D domain, a range of works [12, 63–65] focus on point cloud generation. For more complex surface generation, some works [66, 13, 10, 67, 68, 11] adopt latent diffusion models to learn implicit neural representations and form final shapes via a decoder. For conditional shape completion, both DiffRF [10] and Diffusion-SDF [69] adopt a masked diffusion strategy to fill in missing regions cropped out by 3D boxes. However, their training processes do not leverage a paired incomplete-to-complete ground truth, which may prevent them from accurately learning the completion rules. Contrarily, our method explicitly uses the scan pairs for conditional training, yielding outputs closely resembling the actual objects scanned. Also, we apply the diffusion process in explicit volumetric TSDFs, preserving more geometrical structures during the completing process.

## 3   Method

### 3.1   Formulation

To prepare the training data, we generate an incomplete 3D scan from depth frames using volumetric fusion [21] and represent the scan as a truncated signed distance field (TSDF) in a volumetric grid. Yet, to accurately represent a ground-truth shape, we choose to use a volumetric truncated unsigned distance field (TUDF) instead. This is because retrieving the sign bit from arbitrary 3D CAD models (*e.g.*, some are not closed) is non-trivial. By using TUDF, we can robustly capture the geometric features of different objects without being limited by the topology.

Given such a volume pair, *i.e.*, the incomplete scan $c$ and complete 3D shape $x_0$, we formulate the shape completion as a generation task that produces $x_0$ conditioned on $c$. We employ the probabilistic diffusion model as our generative model. In particular, it contains (i) a *forward process* (denoted as $q(x_{0:T})$) that gradually adds Gaussian noise to corrupt the ground-truth shape $x_0$ into a random noise volume $x_T$, where $T$ is the total number of time steps; and (ii) a *backward process* that employs a shape completion network, with learned parameters $\theta$, to iteratively remove the noise from $x_T$ conditioned on the partial scan $c$ and produce the final shape, denoted as $p_\theta(x_{0:T}, c)$. As both the *forward* and *backward* processes are governed by a discrete-time Markov chain with time steps $\{0, ..., T\}$, their Gaussian transition probabilities can be formulated as

$$q(x_{0:T}) = q(x_0) \prod_{t=1}^{T} q(x_t|x_{t-1}), \quad q(x_t|x_{t-1}) := \mathcal{N}(\sqrt{1-\beta_t}x_{t-1}, \beta_t\mathbf{I}) \tag{1}$$

$$\text{and } p_\theta(x_{0:T}, c) = p(x_T) \prod_{t=1}^{T} p_\theta(x_{t-1}|x_t, c), \quad p_\theta(x_{t-1}|x_t) := \mathcal{N}(\mu_\theta(x_t, t, c), \sigma_t^2\mathbf{I}). \tag{2}$$

In the *forward* process (Eq. (1)), the scalars $\beta_t \in [0, 1]$ control a variance schedule that defines the amount of noise added in each step $t$. In the *backward* process (Eq. (2)), $p(x_T)$ is a Gaussian prior in time step $t$, $\mu_\theta$ represents the mean predicted from our network and $\sigma_t^2$ is the variance. As suggested in DDPM [60], predicting $\mu_\theta(x_t, t, c)$ can be simplified to alternatively predicting $\epsilon_\theta(x_t, t, c)$, which should approximate the noise used to corrupt $x_{t-1}$ in the *forward* process, and $\sigma_t$ can be replaced by the pre-defined $\beta_t$. With these modifications, we can optimize the network parameter $\theta$ with a mean squared error loss to maximize the generation probability $p_\theta(x_0)$. The training objective is

$$\arg \min_\theta E_{t,x_0,\epsilon,c}[||\epsilon - \epsilon_\theta(x_t, t, c)||^2], \quad \epsilon \in \mathcal{N}(0, \mathbf{I}) \tag{3}$$

where $\epsilon$ is the applied noise to corrupt $x_0$ into $x_t$ and $\mathcal{N}(0, \mathbf{I})$ denotes a unit Gaussian distribution. We define all the diffusion processes in a volume space of a resolution $S^3$, *i.e.*, $x_{0:T}, c, \epsilon \in \mathbb{R}^{S \times S \times S}$, where each voxel stores a scalar TSDF/TUDF value; $S$=32 in our experiments. Compared to previous latent diffusion models [11, 13, 66, 10] that require shape embedding first, we directly manipulate the shape with a better preservation of geometric features. Doing so naturally enables our hierarchical feature aggregation strategy (see Sec. 3.2). Next, we will introduce how to predict $\epsilon_\theta(x_t, t, c)$.

### 3.2 Shape Completion Pipeline

**Overview.** To enhance the completion accuracy, we encourage the incomplete scans to control completion behaviors. Inspired from the recent ControlNet [70], which shows great control ability given 2D conditions, we adopt a similar principle to train an additional control branch. To predict the noise $\epsilon_\theta(x_t, t, c)$ in Eq. (3), we encode the corrupted ground-truth shape $x_t$ by a main branch and the partial shape $c$ by a control branch, where two branches have the same network structure without parameter sharing. Owing to the compact shape representation in 3D volume space, both complete and incomplete shapes are encoded into multi-resolution feature volumes with preserved spatial structures. Then, we hierarchically aggregate their features at each network level, as the sizes of two feature volumes are always aligned. In this work, we simply add up the two feature volumes to allow for a spatially-consistent feature aggregation, *i.e.*, only features at the same 3D location are combined. Compared with frequently-used cross-attention technique [11, 13, 57], doing so can significantly reduce computation costs. By multi-scale feature interaction, the network can effectively correlate the difference in partialness between two shapes, both locally and globally, to learn the completion regularities. The final integrated features are then used to predict the noise volume $\epsilon_\theta$.

**Network architecture.** Fig. 2 provides an overview of our network architecture, which has a main branch and a control branch to respectively handle complete and incomplete shapes. The main branch (upper) is a 3D U-Net modified from a 2D version [59]. It takes as input an corrupted complete shape $x_t$, including (i) a pre-processing block $\varepsilon_x(\cdot)$ of two convolution layers to project $x_t$ into a high-dimension space, (ii) $N$ subsequent encoder blocks, denoted as $\{F_x^i(\cdot)\}_{i=1}^{N}$, to progressively encode and downsample the corrupted shape $x_t$ into a collection of multi-scale features, (iii) a middle block $M_x(\cdot)$ with a self-attention layer to incorporate non-local information into the encoded feature volume, and (iv) $N$ decoder blocks $\{D_x^i(\cdot)\}_{i=1}^{N}$ that sequentially upsample features through an inversion convolution to produce a feature volume of the same size as the input $x_t$. The term

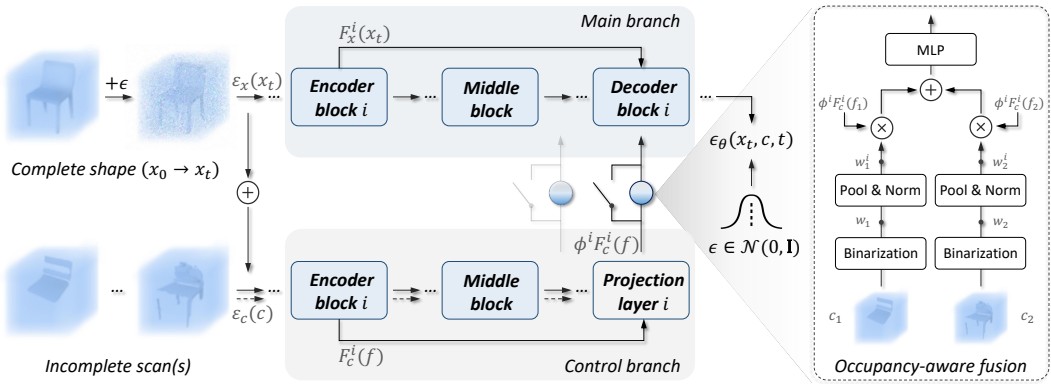

Figure 2: Given a corrupted complete shape $x_t$ (diffused from $x_0$) and an incomplete scan $c$, we first process them into $\varepsilon_x(x_t)$ and $\varepsilon_c(c)$ to align the distributions. We employ a main branch to forward $\varepsilon_x(x_t)$, and a control branch to propagate their fused features $f$ into deep layers. Multi-level features of $f$ are aggregated into the main branch for hierarchical control in predicting the diffusion noise. To support multiple partial scans as condition, *e.g.*, two scans $\{c_1, c_2\}$, we switch on occupancy-aware fusion (Sec. 3.3). This strategy utilizes the occupancy masks to enable a weighted feature fusion for $c_1$ and $c_2$ by considering their geometry reliability before feeding them into the main branch.

"e(d)ncoder/middle block" denotes a group of neural layers commonly adopted as a unit in neural networks, *e.g.*, a "resnet+downsample" block.

We use a control branch (bottom) to extract the features of incomplete scan(s). Likewise, we employ a pre-processing block, $N$ encoder blocks, and a middle block to extract multi-scale features from the conditional input, denoted as $\varepsilon_c(\cdot)$, $\{F_c^i(\cdot)\}_{i=1}^N$, and $M_c$, respectively. They mirror the architecture of the corresponding blocks in the upper branch, yet operating with non-shared parameters, considering the different input data regularities (*e.g.*, complete and sparse). Unlike the upper branch, there are no decoders for computation efficiency. Instead, we append a projection layer after each encoder/middle block $F_c^i/M_c$ to forward multi-scale features and feed them into the decoder blocks of the main branch. As diffusion models are time-conditioned, we also convert the time step $t$ into an embedding via two MLPs and add it to the volume features in each network block (the blue box in Fig. 2).

**Hierarchical feature aggregation.** To integrate the features of the complete and incomplete shapes for conditional control, we fuse them across multiple network levels. In this process, we first consider a *single* incomplete shape (denoted as $c$) as the condition. Given the significant disparity between $c$ and $x_t$ (incomplete v.s. complete, TSDF v.s. TUDF), we employ the pre-processing layers ($\varepsilon_x(\cdot)$ and $\varepsilon_c(\cdot)$) to first empirically project the two different fields into more compatible feature space. Then we fuse them into feature volume $f$, *i.e.*, $f = \varepsilon_x(x_t) + \varepsilon_c(c)$. After that, $f$ is passed through the control branch to better propagate the useful information to the deeper layers. In turn, we use multi-scale condition features from the control branch to guide each level of decoder blocks $\{D_x^i(\cdot)\}_{i=1}^N$ in the main branch. The features that enter the $i$-th decoder block are denoted as

$$d^i = [D_x^{i-1}(x_t), F_x^i(x_t) + \phi^i(F_c^i(f))], \quad f = \varepsilon_x(x_t) + \varepsilon_c(c) \tag{4}$$

where $[\cdot, \cdot]$ is the concatenation operation and $\phi^i$ is one $1 \times 1$ convolution layer for feature projection. For simplicity, we use $F_x^i(x_0)$ to denote the features of the $x_0$ after the $i$-th encoder block $F_x^i(\cdot)$, as $x_0$ is not directly processed by $F_x^i(\cdot)$; the same notation is used for $F_c^i(f)$ and $D_x^{i-1}(x_t)$. The feature after the final decoder block is the network output $\epsilon_\theta(x_t, t, c)$.

By such a design, we can hierarchically incorporate conditional features, leveraging both their local and broader contexts to optimize the network outputs. Interestingly, we observe that adjusting the network level for feature aggregation can alter a trade-off between completion accuracy and diversity. Specifically, if we only aggregate features at the low-resolution network layers, we might miss finer geometry details present in the higher-resolution layers. This could reduce completion accuracy, as the model has less contextual information to work with. In contrast, if aggregating features at all levels, the effective control might make completion results closely resemble the ground truths, yet lowering the diversity. Further ablation study is provided in Sec. 4.5.

### 3.3 Occupancy-aware Fusion

Our framework can also take multiple incomplete scans as inputs. This option not only provides richer information to constrain the completed shape geometry but also enhances the approach's practicality, particularly in scenarios that a single-pass scan may not fully capture the entire object. Our critical design is to effectively fuse multiple partial shape features. As averaging the original TSDF volumes is sensitive to registration noise, we first register multiple partial shapes (*e.g.*, using Fast-Robust-ICP [71]) and propose an occupancy-aware approach to fuse them in the feature space.

Given a set of incomplete shapes of the same object, denoted as $\{c_1, ..., c_M\}$, we individually feed them into the control branch to produce feature volumes $\{F_c^i(f_1), ..., F_c^i(f_M)\}$ after the $i$-th encoder block. Before feeding them into the decoder $D_x^i$, we refer to their occupancy masks to perform a weighted feature average. Concretely, we first compute the original occupancy mask for each partial shape based on the TSDF values, *i.e.*, $w_j = (abs(c_j) < \tau) \in \mathbf{B}^{S \times S \times S}$ for the $j$-th partial shape. $\tau$ is a pre-defined threshold that assigns the volumes near the object surface as occupied and the rest as unoccupied; $\mathbf{B}$ is a binary tensor. Then, we perform a pooling operation to resize $w_j$ into $w_j^i$ to align the resolution with feature volume $F_c^i(f_j)$ and normalize $w_j^i$ by $w_j^i = w_j^i/(w_0^i + .. + w_M^i)$. For the voxels with zero values across all occupancy masks, we uniformly assign them a 1e-2 value to avoid the division-by-zero issue. We rewrite Eq. (4), which is for single partial shape condition, as

$$d^i = [D_x^{i-1}(x_t), F_x^i(x_t) + \psi(\sum_j w_j^i \phi^i(F_c^i(f_j)))], \quad f_j = \varepsilon_x(x_t) + \varepsilon_c(c_j) \tag{5}$$

where $\psi$ is an MLP layer to refine the fused condition features, aiming to mitigate the discrepancies among different partial shape features, as validated in Sec. 4.4. The right part of Fig. 2 illustrates the above process with two incomplete scans as the inputs.

### 3.4 Training and Inference

We first train the network with a single incomplete shape as the conditional input. In this phase, all the network parameters, except the MLP layer $\psi$ for occupancy-aware fusion (in Eq. (5)), are trained with the objective in Eq. (3). When the network converges, we lock the optimized network parameters and efficiently finetune the MLP layer $\psi$ with multiple incomplete shapes as input.

At the inference stage, we randomize a 3D noise volume as $x_T$ from the standard Gaussian distribution. The trained completion networks are then employed for $T$ iterations to produce $x_0$ from $x_T$ conditioned on partial shape(s) $c$. The occupancy-aware fusion is activated only for multi-condition completion. To accelerate the inference process, we adopt a technique from [61] to sub-sample a set of time steps from [1,...,$T$/10] during inference. After obtaining the generated shape volume $x_0$, we extract an explicit 3D mesh using the marching cube algorithm [72].

## 4 Experiment

### 4.1 Experimental Setup

**Benchmarks.** We evaluate on two large-scale shape completion benchmarks: 3D-EPN [14] and PatchComplete [15]. 3D-EPN comprises 25,590 object instances of eight classes in ShapeNet [73]. For each instance, six partial scans of varying completeness are created in the $32^3$ TSDF volumes by virtual scanning; the ground-truth counterpart, represented by $32^3$ TUDF, is obtained by a distance field transform on a 3D scanline method [74]. While using a similar data generation pipeline, PatchComplete emphasizes completing objects of unseen categories. It includes both the synthetic data from ShapeNet [73] and the challenging real data from ScanNet [75]. For a fair comparison, we follow their data splits and evaluation metrics, *i.e.*, mean $l_1$ error on the TUDF predictions across all voxels on 3D-EPN, and $l_1$ Chamfer Distance (CD) and Intersection over Union (IoU) between the predicted and ground-truth shapes on PatchComplete. As these metrics only measure the completion accuracy, we introduce other metrics in Sec. 4.3 to compare multimodal completion characteristics.

**Implementation details.** We first train our network using a single partial scan as input by 200k iterations on four RTX3090 GPUs, taking around two days. If multiple conditions are needed, we finetune project layers $\psi$ for additional 50k iterations. Adam optimizer [76] is employed with a learning rate of $1e^{-4}$ and the batch size is 32. On the 3D-EPN benchmark, we train a specific

Table 1: Quantitative shape completion results on objects of known categories [14].

| $l_1$-err. ($\downarrow$) | Avg. ($\downarrow$) | Chair | Table | Sofa | Lamp | Plane | Car | Dresser | Boat |
|---|---|---|---|---|---|---|---|---|---|
| 3D-EPN [14] | 0.374 | 0.418 | 0.377 | 0.392 | 0.388 | 0.421 | 0.259 | 0.381 | 0.356 |
| SDF-StyleGAN [51] | 0.278 | 0.321 | 0.256 | 0.289 | 0.280 | 0.295 | 0.224 | 0.273 | 0.282 |
| RePaint-3D [78] | 0.266 | 0.289 | 0.264 | 0.266 | 0.268 | 0.302 | 0.214 | 0.285 | 0.243 |
| ConvONet [26] | 0.220 | 0.210 | 0.247 | 0.254 | 0.234 | 0.185 | 0.195 | 0.250 | 0.184 |
| AutoSDF [6] | 0.217 | 0.201 | 0.258 | 0.226 | 0.275 | 0.184 | 0.187 | 0.248 | 0.157 |
| cGCA [79] | 0.185 | 0.174 | 0.212 | 0.179 | 0.239 | 0.170 | 0.161 | 0.204 | 0.143 |
| ShapeFormer [80] | 0.141 | 0.104 | 0.175 | 0.133 | 0.176 | 0.136 | 0.127 | 0.157 | 0.119 |
| PVD [12] | 0.114 | 0.097 | 0.122 | 0.154 | 0.128 | 0.093 | 0.087 | 0.127 | 0.107 |
| PatchComplete [15] | 0.088 | 0.134 | 0.095 | 0.084 | 0.087 | 0.061 | 0.053 | 0.134 | 0.058 |
| DiffComplete (Ours) | **0.053** | **0.070** | **0.073** | **0.061** | **0.059** | **0.015** | **0.025** | **0.086** | **0.031** |

Table 2: Shape completion results on synthetic objects [73] of unseen categories. $\cdot/\cdot$ means CD/IoU.

| CD($\downarrow$)/IoU($\uparrow$) | 3D-EPN [14] | Few-Shot [81] | IF-Nets [49] | Auto-SDF [6] | ConvONet [26] | PatchComplete [15] | Ours |
|---|---|---|---|---|---|---|---|
| Bag | 5.01 / 73.8 | 8.00 / 56.1 | 4.77 / 69.8 | 5.81 / 56.3 | 5.10 / 70.8 | 3.94 / 77.6 | **3.86 / 78.3** |
| Lamp | 8.07 / 47.2 | 15.1 / 25.4 | 5.70 / 50.8 | 6.57 / 39.1 | 5.42 / 52.6 | **4.68** / 56.4 | 4.80 / **57.9** |
| Bathtub | 4.21 / 57.9 | 7.05 / 45.7 | 4.72 / 55.0 | 5.17 / 41.0 | 4.96 / 60.4 | 3.78 / 66.3 | **3.52 / 68.9** |
| Bed | 5.84 / 58.4 | 10.0 / 39.6 | 5.34 / 60.7 | 6.01 / 44.6 | 5.42 / 63.2 | 4.49 / 66.8 | **4.16 / 67.1** |
| Basket | 7.90 / 54.0 | 8.72 / 40.6 | **4.44** / 50.2 | 6.70 / 39.8 | 6.16 / 54.6 | 5.15 / 61.0 | 4.94 / **65.5** |
| Printer | 5.15 / 73.6 | 9.26 / 56.7 | 5.83 / 70.5 | 7.52 / 49.9 | 5.56 / 72.1 | 4.63 / **77.6** | **4.40** / 76.8 |
| Laptop | 3.90 / 62.0 | 10.4 / 31.3 | 6.47 / 58.3 | 4.81 / 51.1 | 4.78 / 57.3 | 3.77 / 63.8 | **3.52 / 67.4** |
| Bench | 4.54 / 48.3 | 8.11 / 27.2 | 5.03 / 49.7 | 4.31 / 39.5 | 4.69 / 49.6 | 3.70 / 53.9 | **3.56 / 58.2** |
| Avg. | 5.58 / 59.4 | 9.58 / 40.3 | 5.29 / 58.1 | 5.86 / 45.2 | 5.26 / 60.1 | 4.27 / 65.4 | **4.10 / 67.5** |

Table 3: Shape completion results on real-world objects [75] of unseen categories. $\cdot/\cdot$ means CD/IoU.

| CD($\downarrow$)/IoU($\uparrow$) | 3D-EPN [14] | Few-Shot [81] | IF-Nets [49] | Auto-SDF [6] | ConvONet [26] | PatchComplete [15] | Ours |
|---|---|---|---|---|---|---|---|
| Bag | 8.83 / 53.7 | 9.10 / 44.9 | 8.96 / 44.2 | 9.30 / 48.7 | 9.12 / 52.5 | 8.23 / **58.3** | **7.05** / 48.5 |
| Lamp | 14.3 / 20.7 | 11.9 / 19.6 | 10.2 / 24.9 | 11.2 / 24.4 | 9.83 / 20.3 | 9.42 / 28.4 | **6.84 / 30.5** |
| Bathtub | 7.56 / 41.0 | 7.77 / 38.2 | 7.19 / 39.5 | 7.84 / 36.6 | 7.93 / 41.2 | **6.77** / 48.0 | 8.22 / **48.5** |
| Bed | 7.76 / 47.8 | 9.07 / 34.9 | 8.24 / 44.9 | 7.91 / 38.0 | 8.14 / 41.6 | 7.24 / **48.4** | **7.20** / 46.6 |
| Basket | 7.74 / 36.5 | 8.02 / 34.3 | 6.74 / 42.7 | 7.54 / 36.1 | 7.39 / 37.0 | **6.60** / 45.5 | 7.42 / **59.2** |
| Printer | 8.36 / 63.0 | 8.30 / 62.2 | 8.28 / 60.7 | 9.66 / 49.9 | 7.62 / 64.9 | 6.84 / 70.5 | **6.36 / 74.5** |
| Avg. | 9.09 / 44.0 | 9.02 / 38.6 | 8.26 / 42.6 | 8.90 / 38.9 | 8.34 / 42.9 | 7.52 / 49.8 | **7.18 / 51.3** |

model for completing shapes of each known category; while on PatchComplete, we merge all object categories to optimize one model for promoting general completion learning. Due to the unknown class IDs at test time, no classifier-guided [56] or classifier-free [77] sampling techniques are used in our diffusion model. On both two benchmarks, all training data are the same across the compared methods for fairness. The truncation distance in TSDF/TUDFs is set as 3 voxel units. More details about network architecture and experiments are available in the supplementary file. Unless otherwise specified, we report the results on *single* partial shape completion.

## 4.2 Main Results

**Completion on known object categories.** On the 3D-EPN benchmark, we compare DiffComplete against SOTA deterministic [14, 15, 51] and probabilistic [6, 78] methods in terms of completion accuracy (*i.e.*, $l_1$ errors). For probabilistic methods, we use the average results from five inferences, each with random initialization, to account for multimodal outcomes. As shown in Table 1 and Fig. 3, DiffComplete improves over state of the arts by 40% on $l_1$ error (0.053 v.s. 0.088), as well as producing more realistic and high-fidelity shapes. Deterministic methods include 3D-EPN [14], ConvONet [26], and PatchComplete [15]. Unlike these learning a one-step map function for shape completion, we iteratively refine the generated shape, thus significantly mitigating the surface artifacts; see visualization comparisons in Fig. 3. Compared to GAN-based SDF-StyleGAN [51] and Autoregressive-based AutoSDF [6], our diffusion-based generative model offers superior mode coverage and sampling quality.

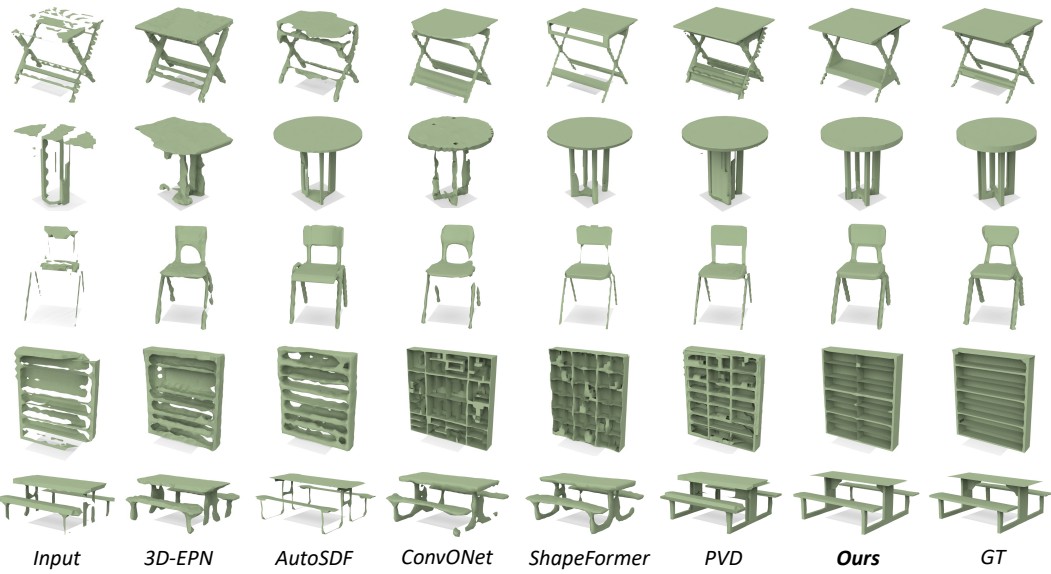

| Input | 3D-EPN | AutoSDF | ConvONet | ShapeFormer | PVD | **Ours** | GT |
|---|---|---|---|---|---|---|---|

Figure 3: Shape Completion on various known object classes. We achieve the best completion quality.

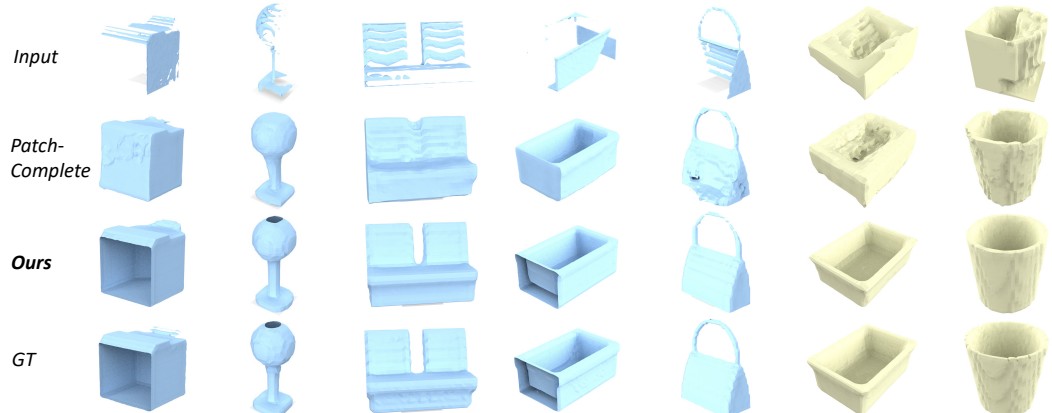

Figure 4: Shape completion on synthetic (blue) and real (yellow) objects of *entirely unseen* classes. Our method produces the completed shapes in superior quality given both synthetic and real data.

Regarding diffusion-based approaches, RePaint-3D is adapted from Repaint [78], a 2D diffusion-based inpainting method that only involves partial shape conditions during the inference process. In contrast, our DiffComplete explicitly matches each partial shape with a complete counterpart at the training stage, thereby improving the output consistency with the ground truths. PVD [12] is originally a point cloud diffusion model. Here, we adapt it to perform TSDF (TUDF) diffusion. A limitation of PVD is that it retains noises of the partial input. Hence, when it is mixed with the generated missing part, noise severely affects the final completion quality. Instead, we design two branches to separately process the partial and complete shapes. By doing so, the model can learn a diffusion process from noise to clean shapes.

**Completion on unseen object categories.** In two datasets of the PatchComplete benchmark, we compare the generalizability of DiffComplete against the state of the arts, including approaches particularly designed for unseen-class completion [81, 15]. As summarized in Table 2, our method exhibits the best completion quality on average for eight unseen object categories in the synthetic ShapeNet data, despite lacking zero-shot designs. The previous SOTA PatchComplete, mainly leverages the multi-scale structural information to improve the completion robustness. Our method inherently embraces this concept within the diffusion models. With our hierarchical feature aggregation, the network learns multi-scale local completion patterns, which could generalize to various object classes,

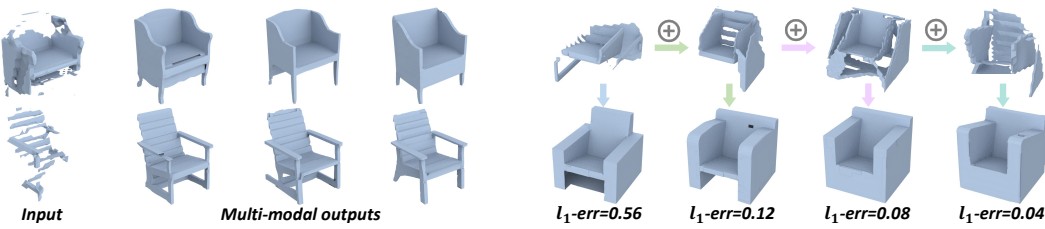

**Input**   **Multi-modal outputs**   $l_1$-err=0.56  $l_1$-err=0.12  $l_1$-err=0.08  $l_1$-err=0.04

Figure 5: Our method produces multimodal plausible results given the same partial shape.

Figure 6: Our method incorporates multiple partial shapes to refine completion results ($l_1$-err. ↓).

Table 4: Multimodal capacity.

| Method | MMD ↓ | TMD ↑ | UHD ↓ |
|---|---|---|---|
| AutoSDF [6] | 0.008 | 0.028 | 0.061 |
| ShapeFormer [80] | 0.007 | 0.024 | 0.055 |
| PVD [12] | 0.007 | 0.027 | 0.042 |
| RePaint-3D [78] | 0.007 | **0.029** | 0.053 |
| Ours | **0.002** | 0.025 | **0.032** |

Figure 7: TMD curve.

Table 5: Multi-condition results.

| Cond. Num. | $l_1$-err. ↓ | MMD ↓ | TMD ↑ |
|---|---|---|---|
| one | 0.07 | 0.002 | **0.025** |
| two | 0.05 | 0.001 | 0.021 |
| three | **0.04** | **0.001** | 0.019 |

Table 6: Effects of fusion choices for multiple conditional inputs.

| Strategy | $l_1$-err. ↓ | CD ↓ | IoU ↑ |
|---|---|---|---|
| simple average | 0.13 | 4.56 | 63.3 |
| w/o MLP $\psi$ | 0.23 | 5.88 | 57.4 |
| occ-aware (Ours) | **0.05** | **3.97** | **68.3** |

Table 7: Feat. aggregation levels.

| 1/8 | 1/4 | 1/2 | 1 | MMD ↓ | TMD ↑ | CD ↓ |
|---|---|---|---|---|---|---|
| ✓ | | | | 0.005 | **0.031** | 4.8 |
| ✓ | ✓ | | | 0.004 | 0.028 | 4.4 |
| ✓ | ✓ | ✓ | | 0.003 | 0.026 | 4.2 |
| ✓ | ✓ | ✓ | ✓ | **0.002** | 0.025 | **4.1** |

Table 8: Ablation on different feature aggregation mechanisms.

| Operation | $l_1$-err. ↓ | MMD ↓ | TMD ↑ |
|---|---|---|---|
| cross-attn. | 0.12 | 0.005 | **0.027** |
| concat. | 0.07 | 0.002 | 0.024 |
| addition (Ours) | **0.07** | **0.002** | 0.025 |

as their local structures are often shared. Our ablation study in Sec. 4.5 further validates this benefit. Table 3 demonstrates our method's superior performance with real-world scans, which are often cluttered and noisy. As showcased in Fig. 4, the 3D shapes produced by DiffComplete stand out for their impressive global coherence and local details.

### 4.3 Multimodal Completion Characteristics

**Quantitative evaluations.** Probabilistic methods have the intriguing ability to produce multiple plausible completions on the same partial shape, known as multimodal completion. For the 3D-EPN chair class, we generate ten results per partial shape with randomized initial noise $x_T$, and employ metrics from prior works [7, 6], *i.e.*, (i) MMD measures the completion accuracy against the ground truths, (ii) TMD for completion diversity, and (iii) UHD for completion fidelity to a partial input. Table 4 shows that our method attains much better completion accuracy and fidelity, while exhibiting moderate diversity. This aligns with our design choice of leveraging the control mechanism to prioritize completion accuracy over diversity. Yet, we can adjust this trade-off to improve shape diversity, as discussed in Sec. 4.5. Fig. 5 presents our multimodal outputs, all showing great realism.

**Effects of input completeness degree.** To verify the influence of input completeness degree on diversity, we select ten chair instances from ShapeNet, due to chair's large structural variation. For each instance, we create six virtual scans of varying completeness, and for each scan, we generate ten diverse completions to compute their TMD. Fig. 7 presents a negative correlation between the shape diversity (reflected by TMD) and input completeness degree, which is measured by the occupied voxel's ratio between the partial and complete GT shapes.

### 4.4 Multiple Conditional Inputs

**Quantitative evaluations.** As indicated in Table 5, DiffComplete consistently improves completion accuracy (lower $l_1$ error) and reduces diversity (lower TMD) when more conditional inputs are added. Fig. 1(b) and 6 showcase the progression of completion results when we gradually introduce more partial shapes of the same object (denoted by the plus symbol). Our model incorporates the local structures of all partial inputs and harmonizes them into a coherent final shape with the lower $l_1$ error.

**Effects of occupancy-aware fusion.** In Table 6, we compare our design with alternatives on the 3D-EPN chair class and PatchComplete benchmark using two conditional inputs. Averaging fea-

tures without occupancy masks largely lowers the completion accuracy due to disturbances from non-informative free-space features. Removing the learnable MLP layer $\psi$ hinders the network's adaptation from single to multiple conditions, also worsening the results. Instead, we adaptively aggregate multi-condition features and refine them to mitigate their discrepancy for reliable completions. Note that directly fusing original scans in TSDF (TUDF) space might be vulnerable to registration errors, as compared with our strategy in the supplementary file.

### 4.5 Ablation Study

**Effects of hierarchical feature aggregation.** Table 7 shows the effects of aggregating features of the complete and incomplete shapes at different decoder levels (see Eq. (4)). First, increasing feature aggregation layers (from single to hierarchical) consistently boosts the completion accuracy (lower MMD), while decreasing it yields better diversity (higher TMD). If connecting features only at the network layer with 1/8 resolution, we achieve the best TMD that surpasses other methods (see Table 4). Thus, the accuracy-diversity trade-off can be adjusted by altering the level of feature aggregation. Second, hierarchical feature aggregation facilitates unseen-class completion (lower CD, tested on PatchComplete benchmark). This improvement suggests that leveraging multi-scale structural information from partial inputs enhances the completion robustness.

**Effects of feature aggregation mechanism.** We ablate on the way to aggregate $F_x^i(x_t)$ and $\phi^i(F_c^i(f)$ in Eq. (4). As Table 8 shows, direct addition achieves the best completion accuracy (the lowest $l_1$-err and MMD), as it only combines features at the same 3D location to precisely correlate their difference for completion. Cross-attention disrupts this spatial consistency and yields less accurate results. Concatenation has a similar performance with addition, while the latter is more efficient.

## 5 Conclusion and Discussion

We presented DiffComplete, a new diffusion-based approach to enable multimodal, realistic, and high-fidelity 3D shape completion, surpassing prior approaches on completion accuracy and quality. This success is attributed to two key designs: a hierarchical feature aggregation mechanism for effective conditional control and an occupancy-aware fusion strategy to seamlessly incorporate additional inputs for geometry refinement. Also, DiffComplete exhibits robust generalization to unseen object classes for both synthetic and real data, and allows an adjustable balance between completion diversity and accuracy to suit specific needs. These features position DiffComplete as a powerful tool in various applications in 3D perception and content creation. Yet, as with most diffusion models, DiffComplete requires additional computation due to its multi-step inference. A comprehensive discussion on the limitation and broader impact is provided in the supplementary file.

## Acknowledgement

This work is partially supported by the Research Grants Council under the Areas of Excellence scheme grant AoE/E-601/22-R, Shenzhen Science and Technology Program KQTD20210811090149095, and the Research Grants Council of the Hong Kong Special Administrative Region, China (CUHK 14206320).

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
