# Supplementary Material of DiffComplete: Diffusion-based Generative 3D Shape Completion

## Contents

## A   Detailed Network Architecture

Fig. 8 shows the detailed architecture of encoder blocks, middle blocks, and decoder blocks of our network (corresponding to those in Fig. 2). In particular, both the main and control branches consist of four encoder blocks (Fig. 8(a)), built from repeated ResBlocks and Downsample layers, where the latter iteratively reduces the feature volume to 1/8th of its original size. The middle block (Fig. 8(b)) comprises two ResBlocks with an intermediate AttentionBlock. The main branch additionally contains four decoder blocks (Fig. 8(c)), which restore the volume shape to its initial size using upsampling. Fig. 8(d) presents the detailed structure of a ResBlock unit. It receives features from the preceding network layer and a time embedding as inputs, fuses their embeddings, and processes them with convolutional operations. To support network and experiment reproduction, we will make our code available.

## B   Additional Experiments

### B.1   Choice of Training Strategy

In Table 9, we evaluate different training strategies for our model on chair class of the 3D-EPN [14] benchmark. Both 'pretraining-class' and 'pretraining-all' follow ControlNet [70]'s training paradigm. They involve an initial unconditional generation task for main branch pretraining and then optimize only the control branch with partial shapes with the objective in Eq. (3). Specifically, 'pretraining-class' pretrains the main branch on individual object classes, while 'pretraining-all' employs all data for one pretraining model, both of which are finetuned on a specific object class. These pretraining-based strategies bring much more completion errors (higher $l_1$-err.)  and increased completion

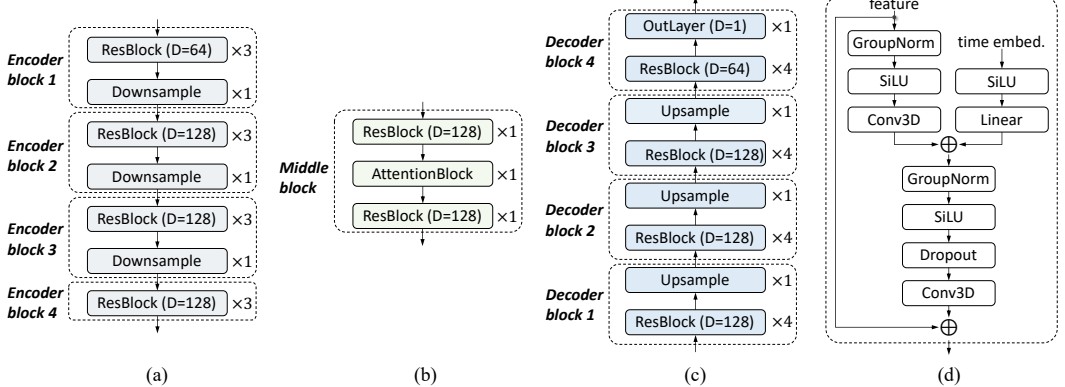

Figure 8: The detailed architecture of encoder blocks (a), middle blocks (b), and decoder blocks (c) in our network, each is mainly constructed by stacked ResBlocks (d). The 'D' denotes the output feature dimension and $\oplus$ represents the feature addition operation.

Table 9: Choice of training strategy. The pretraining-based options increase the completion errors ($l_1$-err.) and diversity (TMD).

| Training Strategy | $l_1$-err. ↓ | TMD ↑ |
|---|---|---|
| pretraining-class | 0.14 | 0.028 |
| pretraining-all | 0.11 | **0.030** |
| scratch (Ours) | **0.07** | 0.025 |

Table 10: Choice of fusion space for multiple partial shapes. Directly fusing them in the original TSDF space significantly impairs the completion quality.

| Space | Strategy | $l_1$-err. ↓ | CD ↓ | IoU ↑ |
|---|---|---|---|---|
| TSDF | simple average | 0.29 | 6.68 | 51.6 |
| | occ-aware | 0.12 | 4.78 | 61.0 |
| feature | simple average | 0.13 | 4.56 | 63.3 |
| | occ-aware | **0.05** | **3.97** | **68.3** |

diversity (higher TMD). This may be due to the model's over-relies on the learned distribution during pretraining, making it less adaptable to concrete completion tasks. Instead, we train the network entirely from scratch, which is most effective for accurate shape completion (with the lowest $l_1$-err.).

### B.2 Choice of Fusion Space for Multiple Conditions

Table 10 analyzes the impacts of fusion space when incorporating multiple incomplete shapes as conditions, which supplements Table 6 to further validate our fusion choice. The experiments are conducted on the 3D-EPN [14] and Patchcomplete [15] benchmarks. The first two rows refer to merging multiple aligned partial shapes into a volumetric TSDF and feeding it to the network as a single condition. However, both 'simple average' and 'occupancy-aware' fusion strategies for this yield worse performance across diverse metrics. This is because registration errors between different shapes directly disrupt the input, potentially causing distortions in the final completed model. In contrast, we move the fusion process to a more abstract level within the hierarchical feature space, which can be more resilient to simple noise at the TSDF level. Our occupancy-aware fusion strategy further provides more accurate and robust completion results.

### B.3 Choice of Occupancy Threshold

During the occupancy-aware fusion process, the TSDF value threshold $\tau$ determines which volumes are recognized as occupied. In Table 11, we evaluate the impact of different thresholds on the completion performance using the 3D-EPN [14] and Patchcomplete [15] benchmarks. Selecting extreme threshold values, whether very low (*e.g.*, 1 voxel unit) or high (*e.g.*, 5 voxel units), tends to degrade results, as lower thresholds may omit informative geometries while higher ones could include redundant geometries that do not contribute meaningfully to the object shape, both of which confuse the model. Conversely, a middle-range value (3 voxel unit) provides a balance between preserving essential geometries and avoiding unnecessary ones, thereby achieving optimal completion accuracy (the lowest $l1$-err. and CD).

Table 11: Choice of occupancy threshold. Extremely low or high values yield worse results.

| Threshold $\tau$ | $l_1$-err. $\downarrow$ | CD $\downarrow$ | IoU $\uparrow$ |
|---|---|---|---|
| 1 | 0.08 | 4.21 | 66.6 |
| 2 | 0.05 | 4.09 | 68.2 |
| 3 (Ours) | **0.05** | **3.97** | 68.3 |
| 4 | 0.06 | 4.10 | **68.4** |
| 5 | 0.07 | 4.13 | 67.8 |

Figure 9: Accuracy (MMD) and diversity (TMD) curves with varying training iterations.

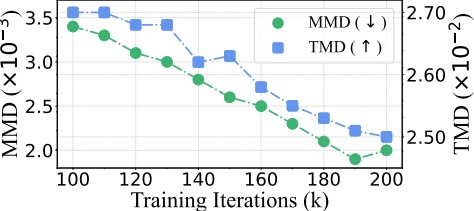

Table 12: Finetuning on eight unseen object categories with limited data. Our model shows substantial improvements with incremental data. The 0% and 100% indicate zero and full data usage, respectively.

| Data Ratio | Avg. CD $\downarrow$ | Bag | Lamp | Bathtub | Bed | Basket | Printer | Laptop | Bench |
|---|---|---|---|---|---|---|---|---|---|
| 0% | 3.19 | 2.98 | 3.54 | 2.87 | 3.24 | 3.70 | 3.46 | 2.85 | 2.87 |
| 1% | 2.84 | 2.54 | 3.27 | 2.46 | 2.98 | 3.24 | 3.25 | 2.49 | 2.52 |
| 5% | 1.84 | 1.67 | 1.93 | 1.63 | 2.01 | 1.96 | 2.04 | 1.81 | 1.70 |
| 10% | 1.34 | 1.23 | 1.43 | 1.20 | 1.40 | 1.49 | 1.48 | 1.28 | 1.19 |
| 100% | 1.02 | 0.98 | 1.07 | 0.95 | 1.05 | 1.12 | 1.10 | 1.00 | 0.95 |

## B.4  Impact of Training Iteration

Fig. 9 plots the performance of our model over a range of training iterations from 100k to 200k. As the training iterations increase, the shape completion accuracy improves (with lower MMD) while the completion diversity gradually decreases (with lower TMD). This trend reveals that, given more training time, the model learns to better fit the target distribution.

## B.5  Data-efficient Finetuning on Unseen Categories

In Table 12, we evaluate the model's ability to complete ShapeNet objects [73] of unknown categories when finetuned with limited data. To this end, we first divide the data from unseen classes into a 7:3 train-test split. Then we finetune our model using varying proportions of the training set (1%, 5%, and 10%). Here, a ratio of 0% indicates no finetuning process, following the setting in Table. 2, while 100% means using the entire training set. A lower CD denotes better completion accuracy.

With just 1% finetuning data, the average CD decreases by 10.9% (from 3.19 to 2.84). A more substantial improvement is observed when the data ratio increases to 5%, with a nearly 1 point decrease in average CD compared to the 1% ratio. The trend of improvement continues for a 10% data ratio, where the model impressively approaches the performance achieved using the full training set. These results demonstrate that our model has a robust few-shot learning capability and can generalize well from a small amount of out-of-distribution data.

## B.6  Comparison with Point Cloud Completion Approaches

Regarding methods on point cloud completion, we compare our DiffComplete with the leading approach SnowflakeNet [82]. To make the comparisons fair, we converted SnowflakeNet's outputs to meshes using the reconstruction technique from ConvONet [26]. On the 3D-EPN benchmark, SnowflakeNet achieves the average $l_1$-error ($\downarrow$) of 0.189 across eight object classes, while our method attains the 0.053 $l_1$-error, which delivers much better completion accuracy.

## B.7  Applications in Semantic Part Editing

As our methods flexibly supports multiple conditional inputs, it can be applied for editing a semantic part of the object from one or multiple partial inputs. Fig. 10 showcases the visual examples for the shape editing task to better demonstrate the model's multiple-input capability.

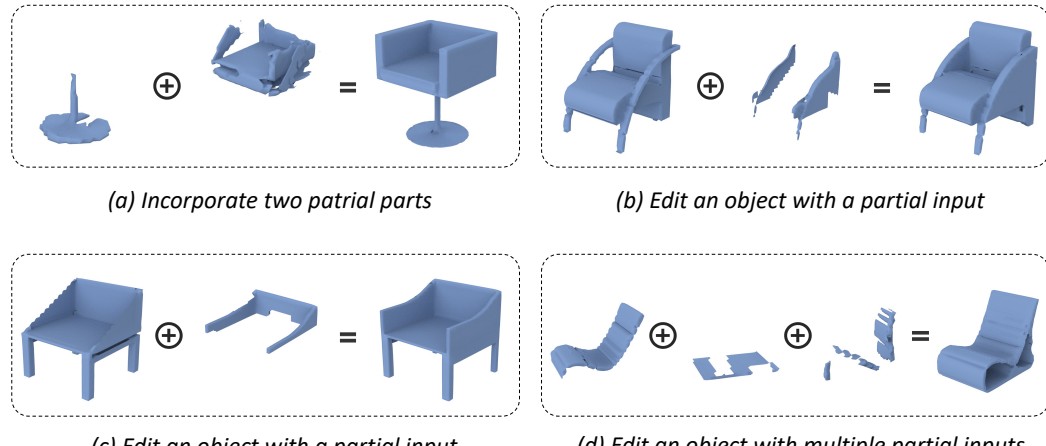

*(a) Incorporate two patrial parts*

*(b) Edit an object with a partial input*

*(c) Edit an object with a partial input*

*(d) Edit an object with multiple partial inputs*

Figure 10: Examples of multi-input shape completion for part editing. (a) Incorporate two distinct partial parts to complete an object. (b)-(c) Edit a specific part of the object using a partial input. It is achieved by directly taking an object shape and an incomplete part as the conditional inputs. (d) Edit an object with multiple partial inputs.

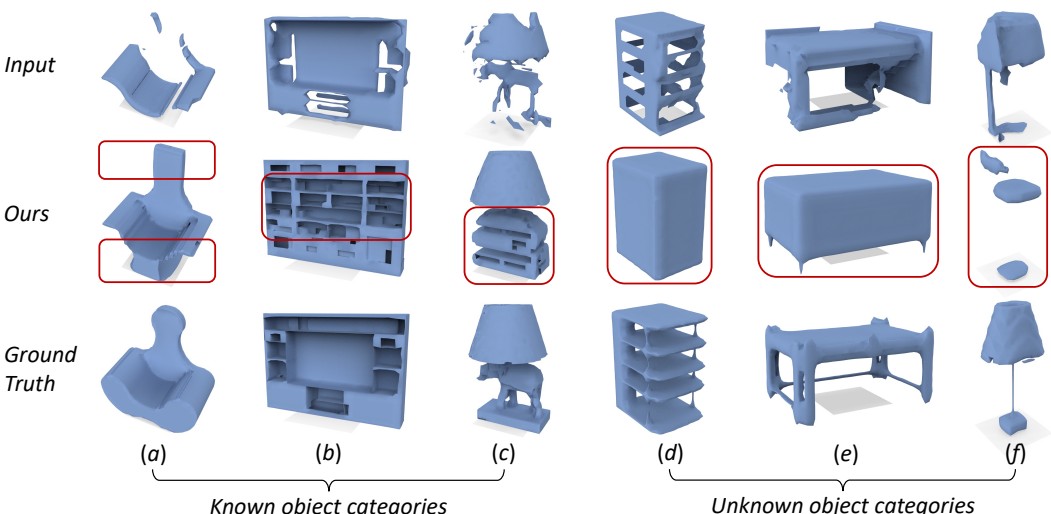

Figure 11: Failure cases on known and unknown object categories. Our model may produce inaccurate or improbable completions when faced with overly sparse inputs (a), atypical shapes (b-c), complex structures (d), and high noise levels (e-f). The red boxes highlight the difference with ground truths.

### B.8 Failure Cases

Fig. 11 showcases certain failure instances in our completion results. For shape completion on known object categories (a-c), given the overly sparse input, our model struggles with producing a shape that aligns with the ground truth (see (a) and (b)). In example (c), the model fails to complete a non-standard structure, such as an elephant beneath a lamp. This can be attributed to the model's tendency to generate shapes based on frequently seen patterns during training, while the elephant belongs to an atypical structure in lamp categories. For unknown object categories (d-f), the model faces additional challenges. The case (d) reveals that our model may favor simple structures when the input shape is too complex. Cases (e) and (f) further show the model's difficulty in handling substantial input noise, which results in inaccurate or improbable completions.

## C   Quantitative Visualizations

**Visualizations on known object categories.** Fig. 12 shows quantitative results on diverse object categories produced by SOTA PatchComplete and our DiffComplete. Our method produces the completion results with much fewer artifacts and more realistic shapes. Our completions are also highly accurate, closely recovering the ground-truth shapes.

**Visualizations on unseen object categories.** As shown in Fig. 13, across a diverse set of entirely unseen object categories, our method also achieves superior completion results over PatchComplete, preserving better global coherence and local details. Note that we do not employ any zero-shot designs while PatchComplete does.

**Multimodal completion results.** In Fig. 14, we show multiple plausible completion results produced by our model from the same partial shape input. For sparser input shapes, our model can explore different possibilities to fill in the missing regions and yield more diverse results (*e.g.*, the first row). In contrast, for the inputs with higher completeness levels, our control-based design ensures the model to output more consistent 3D shapes (*e.g.*, the last row).

**Visualizations of denoising process.** In Fig. 15, we visualize the produced shapes at different time steps during the inference stage. Our model progressively converts the noises into clean 3D shapes.

## D   Difference from ControlNet [70]

Although the paradigm of injecting conditional features takes inspiration from ControlNet, DiffComplete has critical distinctions. First, DiffComplete differs from ControlNet in several key areas. (i) Task: ControlNet tackles the 2D text-to-image generation task, making it work on our 3D completion task is non-trivial. To this end, we design the appropriate volumetric representation and 3D networks. (ii) Motivation: The motivation of ControlNet is to finetune pretrained large diffusion models, while we aim to train a specific diffusion model. This leads to different training strategies. (iii) Training Strategy: ControlNet utilizes a "trainable copy" initialization, but our experiments found that training from scratch is the most effective way, as shown in Table 9. (iv) Design Details: We discard "zero convolution", a critical component in ControlNet, as we do not require the pre-training process. We also directly embed the original 3D shape representation rather than operating in latent space.

Second, DiffComplete offers new features and insights beyond ControlNet. (i) Our method further supports multiple inputs to improve the completion accuracy. (ii) We delve deeper into the feature injection mechanism, observing that altering the feature aggregation level finely controls the trade-off between completion accuracy and diversity. This finding can be leveraged to adjust the model's performance, as described in Sec. 4.5 of the main paper.

## E   Limitations and Future Work

First, our model may struggle to complete highly irregular or noisy shapes, as extensively discussed in Sec. B.8. Yet, our model's multimodal capacity will increase the likelihood of producing satisfactory results, enabling it to tackle this problem more effectively than deterministic methods. With more diverse training data, the model's performance on completing these hard shapes could be improved.

Like most diffusion models [60, 59], another limitation of DiffComplete is the substantial computational requirements due to the iterative completion process. Despite employing the technique in work [61] to reduce sampling steps by ten times, we still need 100 steps to achieve competitive results, which costs around 3 4 seconds per shape on an RTX 3090 GPU. The extended test time could cap its potential for real-time or resource-constrained applications. Future work will leverage the advances in fast sampling techniques (*e.g.*, [83]) to accelerate inference speed.

Also, the dense 3D CNN architecture in our implementation limits the model ability to handle high-resolution 3D shapes, due to the cubic increase in computational costs with volume size. A potential solution could be replacing dense CNNs with efficient 3D network modules, such as SparseConv [84], TriVol [85], or Octree-based layers [86], while remaining other essential designs of our framework. Decomposing 3D shapes into parts can also help the model capture high-resolution details with low memory costs. For instance, we can utilize the techniques of work [69] to first encode global volumetric TSDF (TUDF) into smaller structural patches. Then, our DiffComplete is capable of performing accurate part-level completion. Such an adaptation is feasible due to the versatility of DiffComplete's core designs. Future work will explore more efficient 3D representations (e.g., patch-based) to enhance the capability of our method.

At last, although our model shows robust generalizability to unseen object classes, its performance may be adversely affected by the quality and diversity of the training data. In cases where object classes or shapes deviate significantly from the training set, the model may underperform. Therefore, careful selection of training data is needed to boost completion robustness.

In conclusion, despite the current limitations, they also present opportunities for model improvement. By addressing these issues, we believe the full potential of our model can be further realized.

## F   Broader Impact

On the positive side, the potential applications of our work are widespread. DiffComplete could contribute to fields such as computer vision, robotics, virtual reality, and many others. For instance, in computer vision and robotics, our method can significantly enhance object reconstruction capabilities, providing more accurate and realistic models that facilitate object recognition, manipulation, and robot navigation. Similarly, in virtual reality or 3D printing, our model is able to complete or refine 3D models, enriching the user experience and the quality of end products.

Moreover, our model provides a flexible balance between the completion diversity and accuracy. This attribute can be tailored to suit various application needs, thereby broadening its potential usability across different tasks.

On the other hand, it is crucial to consider potential negative implications. As with any AI technology, there are risks associated with misuse. For instance, if used for recreating personal items without consent, it could lead to unwarranted privacy intrusions. In addition, the automation facilitated by our model may also displace jobs involving manual 3D modeling or shape completion.

To conclude, while our research holds promising potential, it is essential to responsibly manage its broader impacts. We advocate for developing this technology in a way that maximizes societal benefits and minimizes potential negative effects.

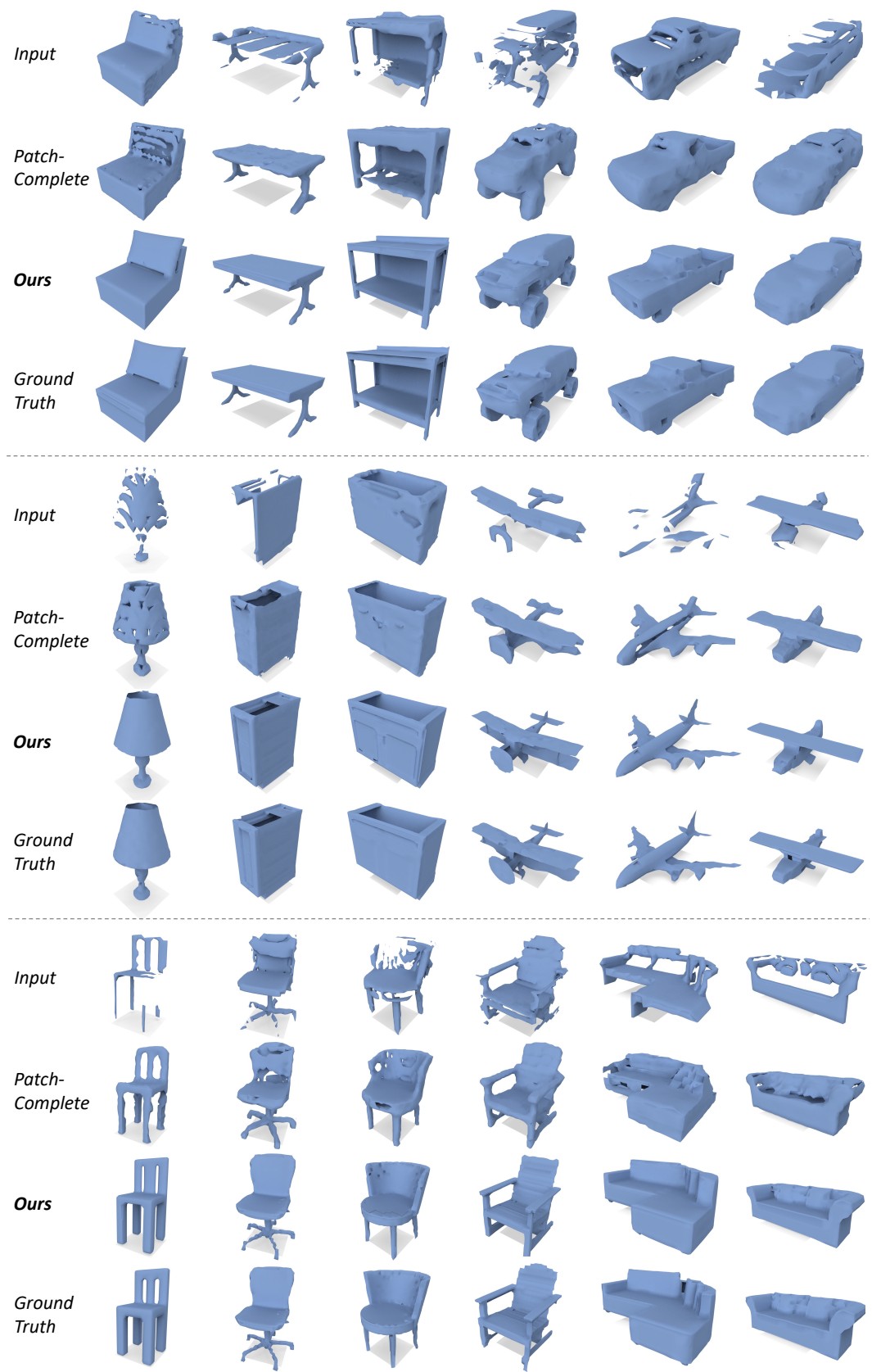

Figure 12: Quantitative results on completing objects of diverse known categories. Our method significantly outperforms SOTA PatchComplete [15] on both the completion quality and accuracy.

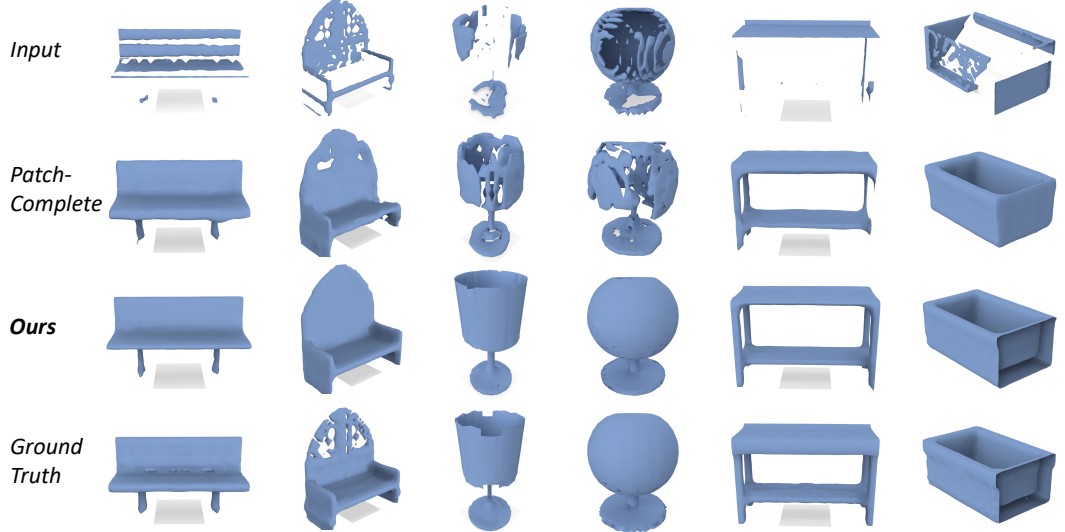

Figure 13: Quantitative results on completing objects of entirely unseen categories. Our significantly outperforms SOTA PatchComplete [15] on both the completion quality and accuracy.

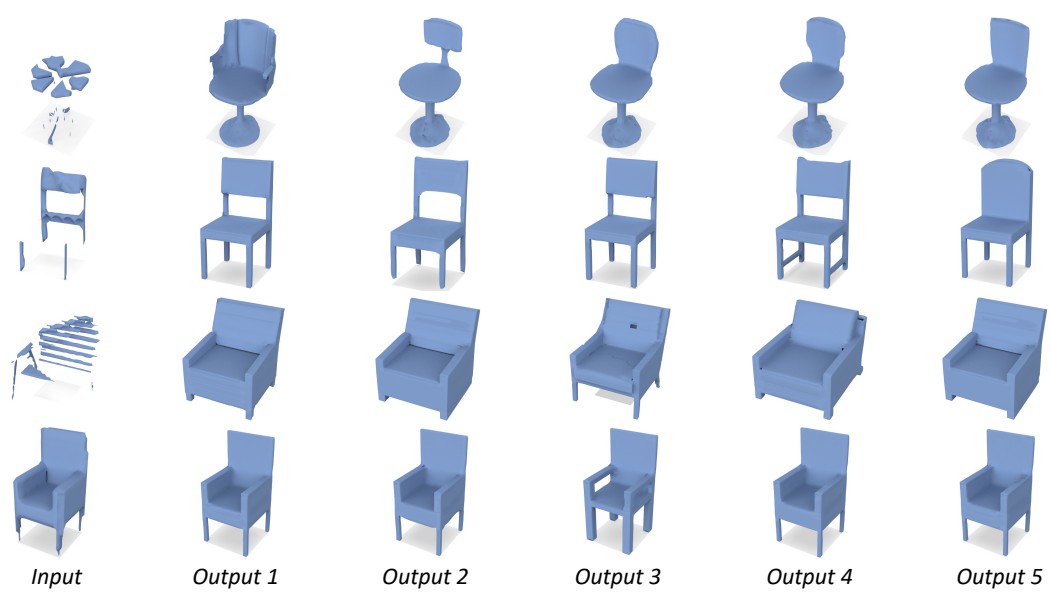

Figure 14: Multimodal completion results on ShapeNet Chair class. We run the model five times for the same input. The level of input sparsity affects the diversity and certainty of the shape completion.

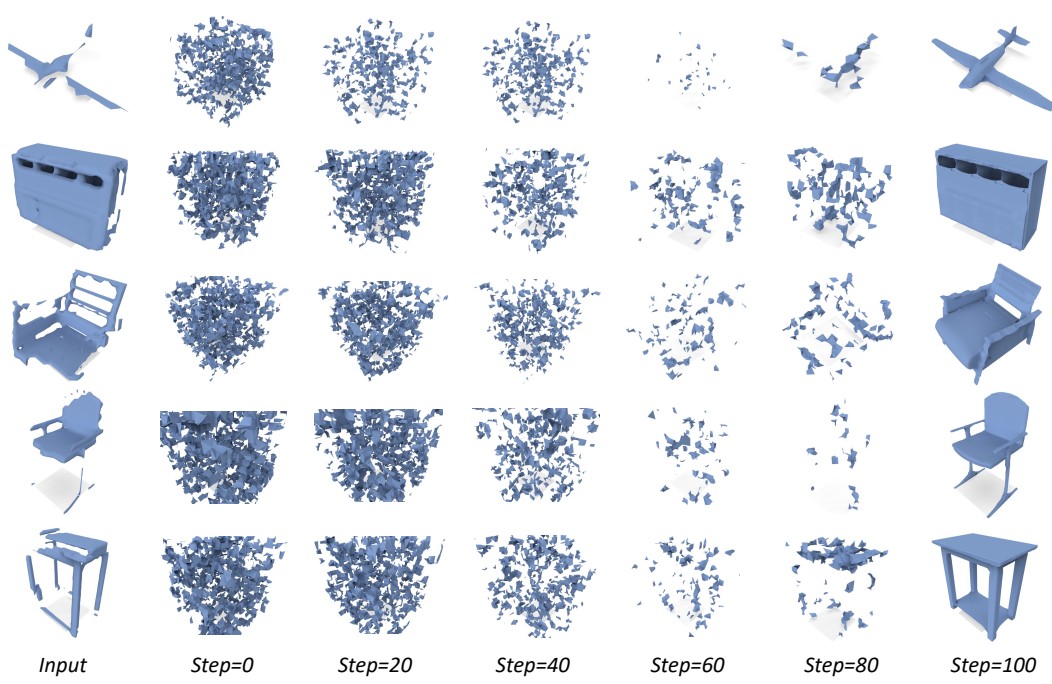

| Input | Step=0 | Step=20 | Step=40 | Step=60 | Step=80 | Step=100 |

Figure 15: The denoising process that gradually converts the noises to completed shapes (from left to right). We visualize the produced shapes at varying time steps (0, 20, 40, 60, 80, and 100).