# OpenReview forum: "DiffComplete: Diffusion-based Generative 3D Shape Completion"
_NeurIPS.cc/2023/Conference — NeurIPS 2023 poster_

### Official Review · Reviewer_9z1Q · 2023-07-03

**Soundness:** 4 excellent
**Presentation:** 4 excellent
**Contribution:** 3 good
**Rating:** 8
**Confidence:** 5

**Summary:**

The paper tackles the shape completion task by introducing a diffusion-based technique. To do so, additional design choices have been made such as hierarchical feature aggregation and an occupancy-aware fusion strategy to better reproduce the shape and respect the details. Based on numerous quantitative and qualitative experiments, the proposed method outperforms others by a considerable margin.

**Strengths:**

The paper clearly outperforms other SOTA.
The visual quality of the completion is good.
Occupancy-aware fusion and hierarchical feature aggregation that are the main task specific designs make sense and the authros have successfully shown their importance in the design through ablation studies.
I like the adoption of control net in the design. I believe it makes sense to learn gradually and at different scales.
The fact that the data is from the actual scan and it is not synthesized by adding noise and removing parts make the method very useful and interesting.


**Weaknesses:**

In terms of writing, the paper could use simpler and more understandable and short sentences. For example, this sentence could be improved by making it shorter or breaking it into two sentences:
To improve the completion accuracy, we enable effective control from single-input condition by hierarchical and spatially-consistent feature aggregation, and from multiple ones by an occupancy-aware fusion strategy.
This happens many times throughout the paper.
Missing reference:
Multimodal shape completion with IMLE:
https://openaccess.thecvf.com/content/CVPR2022W/DLGC/html/Arora_Multimodal_Shape_Completion_via_Implicit_Maximum_Likelihood_Estimation_CVPRW_2022_paper.html

**Questions:**

When you pro-process the input and the incomplete shape through a convolutional network and add them, why do you call it aligning distributions?
As the computation and time are two limitations of this method, would a part-based completion task be a better approach? In fact, my question is whether a part based approach in which a model is first broken into parts will save computation time and understand the geometry better?

**Limitations:**

The method is mostly trained category specific. Considering the time and computational cost of this approach, this makes it hard and expensive to be adopted and modified. easily. However, I weigh the quality of the results more than the time and computation so I am okay with this limitation.

---

> ### Author Rebuttal · Authors · 2023-08-08
>
> Dear Reviewer 9z1Q,
>
> Thank you for the valuable comments and encouraging feedback. We are happy to respond to each question below.
>
> **Q1: The paper could use simpler, more understandable, and short sentences.**
>
> A1: In the revision, we will carefully polish the manuscript to ensure better clarity and conciseness. For instance, the suggested example will be rephrased as "We enhance the completion accuracy with two key designs. For a single condition, we propose a hierarchical feature aggregation strategy to effectively control the outputs. For multiple conditions, we introduce an occupancy-aware fusion strategy to incorporate more shape details."
>
> **Q2: Add reference: Multimodal Shape Completion with Implicit Maximum Likelihood Estimation.**
>
> A2: Both IMLE and DiffComplete address multimodal shape completion, and we'll cite this paper in the revision. The relevant discussion will be inserted in line 102 of the paper and is shown as follows, with the new contents in bold.
>
> "While some generative models can generate diverse global shapes given a partial input, they potentially allow for a high generation freedom and overlook the completion accuracy. **IMLE, for instance, adopts an Implicit Maximum Likelihood Estimation technique to specially enhance structural variance among the generated shapes.** Distinctively, we formulate a diffusion model for shape completion. Our method mainly prioritizes fidelity relative to ground truths while preserving output diversity."
>
> **Q3: Why do you call the pre-processing process of incomplete and complete shapes "aligning distributions"?**
>
> A3: The complete and incomplete shapes are represented by different data fields, *i.e.*, TUDF and TSDF, respectively. Directly fusing them could result in confusing representations. Hence, we use a pre-processing layer to empirically project the two different fields into a more compatible feature space for interaction. We previously referred to this as "aligning distributions". To avoid any misunderstanding, we will remove the term "aligning distributions" and incorporate the above explanations in the revision.
>
> **Q4: Will a part-based approach save computation time and help understand the geometry better?**
>
> A4: Decomposing 3D shapes into parts can help the model capture high-resolution details with low memory costs. We agree that this strategy is beneficial to our shape completion task. For instance, we can utilize the techniques of work [1] and [2] to first encode global volumetric TSDF (TUDF) into smaller structural patches. Then, our DiffComplete is capable of performing accurate part-level completion. Such an adaptation is feasible due to the versatility of DiffComplete's core designs. Future work will explore more efficient 3D representations (*e.g.,* patch-based) to enhance the capability of our method. We'll add these discussions to the future work section in our revised paper.
> ****
> **References**
>
> [1] Li, et al. Diffusion-SDF: Text-to-Shape via Voxelized Diffusion. In CVPR 2023.
>
> [2] Hertz, et al. SPAGHETTI: Editing Implicit Shapes Through Part Aware Generation. TOG 2022.

---

> > ### Comment · Reviewer_9z1Q · 2023-08-18
> >
> > I am still positive about this paper and it should be accepted.

---

> > > ### Author Response · Authors · 2023-08-18
> > > **Author Response to Reviewer 9z1Q**
> > >
> > > Dear Reviewer 9z1Q,
> > >
> > > We sincerely appreciate your positive feedback and consistent support. Your suggestions are instrumental in enhancing the quality of our paper.

---

### Official Review · Reviewer_hw6P · 2023-07-04

**Soundness:** 3 good
**Presentation:** 2 fair
**Contribution:** 3 good
**Rating:** 5
**Confidence:** 4

**Summary:**

The authors present a diffusion-based neural network for 3D shape completion. They introduce a hierarchical and spatially consistent feature aggregation scheme to fuse partial and complete shapes, and an occupancy-aware fusion strategy to fuse multiple partial shapes. They demonstrate a new SOTA performance in shape completion for both new instances and completely unknown object categories.

**Strengths:**

1. The authors successfully apply the diffusion-based model to the shape completion task and achieve a new SOTA performance.

2. The stepwise addition of partial scans to predict a complete shape is interesting and makes completion tasks interactive and adaptable.

3. The proposed model supports multimodal completions that can predict multiple plausible results.

**Weaknesses:**

1. The writing of this paper can be improved.

2. The technical designs, such as the two feature aggregation schemes, are not well motivated. As a main contribution, I expect more explanations of the design ideas.

3. The experimental design is not very clear. Does DiffComplete train on a single category? Since PatchComplete trains on all categories, DiffComplete should also train across categories to allow a fair comparison.

4. The results of multiple conditional inputs are unpredictable. A better example is adding and editing a semantic part of the object from a partial input.

5. Does the proposed model require much more time than its competitors?

**Questions:**

1. I wonder if the resolution can go up to 64^3 or even 128^3?

2. The diversity of multimodal complement is worse than other methods. So is it because the diffusion model gives more deterministic results than others?

**Limitations:**

The limitation is not addressed.

---

> ### Author Rebuttal · Authors · 2023-08-08
>
> Dear Reviewer hw6P,
>
> Thanks for the constructive comments. Please find our response to the specific points below.
>
> **Q1: Explain more design ideas about the two feature aggregation schemes.**
>
> A1: Our key design motivation is to enhance the completion accuracy and generalization ability. Then, we illustrate how each design choice benefits the goals.
>
> **(i) Two-branch architecture**: We utilize two branches to separately process the complete and incomplete scans, such that each branch has a specific focus. Given the same architecture of the two branches, the feature sizes of complete and incomplete shapes are always aligned. This alignment greatly simplifies the following feature aggregation process.
>
> **(ii) Hierarchical feature aggregation**: Leveraging the multilayer structure of CNNs, we aggregate two-branch features at various network levels. It brings two-fold benefits. First, the network can correlate the difference between two shapes at multiple abstraction levels. The learned completion regularities are multi-scale, thus being more accurate and robust. As the substructures of objects are more general than full shapes, *e.g.*, chairs and tables may share very similar legs, the learned local completion pattern could generalize to various object classes. Second, we found that altering the level of feature aggregation can control the trade-off between completion accuracy and diversity. Hence, we can leverage this feature to adjust the model's performance, as illustrated in Table 7 of the main paper.
>
> **(iii) Spatially-consistent feature aggregation**: We simply add up the two-branch features at the respective 3D locations. This operation precisely compares the corresponding parts of two scan pairs, contributing to yield more accurate completion than the widely-adopted cross-attention mechanism, as shown in Table 8 of the main paper.
>
> We'll further clarify our design motivation in the revision.
>
> **Q2: Does DiffComplete train on a single category? Since PatchComplete trains on all categories, DiffComplete should also train across categories to allow a fair comparison.**
>
> A2: For fair comparisons, on the PatchComplete benchmark, we train DiffComplete across all object categories like PatchComplete. This has been described in lines 252-254 of the main paper. On the 3D-EPN benchmark, both PatchComplete and our method train a specific model for each object category. We will make the experimental settings more clear in the revision.
>
> **Q3: To show the results of multiple conditional inputs, a better example is adding and editing a semantic part of the object from a partial input.**
>
> A3: We have provided several visual examples in the attached **PDF** under *the response to all reviewers*. These examples will be integrated into our revised manuscript to better showcase the model's multiple-input capability.
>
> **Q4: Does the proposed model require much more time than its competitors?**
>
> A4: As our method belongs to the class of diffusion model, like other diffusion-based approaches, it's inherently slower than methods of other classes (*e.g.*, GAN) due to the iterative sampling process. In scenarios where the completion quality weighs more than inference time, DiffComplete is definitely a better choice. Additionally, we'd like to mention that the inference time can be reduced by employing fast sampling techniques (*e.g.*, [1]) for diffusion models.
>
> Compared with two other diffusion models [2] and [3], our method maintains similar inference times but significantly improves the completion quality, as shown in **Table R8**.
>
> **Table R8:** Comparisons of completion accuracy (Avg. $l_1$-err.) and average inference time on the 3D-EPN benchmark. It is tested on a single RTX 3090 GPU with a batch size of 1.
> | **Method**|**Paradigm**|**Avg. $l_1$-err. $\downarrow$**|**Inference Time $\downarrow$**|
> |-|-|:-:|:-:|
> |RePaint-3D [2]|diffusion model|0.374|34.1 s|
> |PVD [3]|diffusion model|0.114|**3.1 s**|
> |DiffComplete (Ours)|diffusion model|**0.053**|3.2 s|
>
>
> **Q5: Can the resolution go up to $64^3$ or even $128^3$?**
>
> A5: Yes. **(i)** It's feasible to train our DiffComplete with a resolution of $64^3$ on affordable NVIDIA 3090 GPUs. We provide visual examples of our outputs in the attached **PDF** under *the response to all reviewers*. We are happy to include these results in the supplement during revision.
>
> **(ii)** There are various available strategies to scale up resolutions further (*e.g.*, to $128^3$) even on smaller GPUs. For instance, we can apply the gradient accumulation technique to break the large batch into smaller chunks, ensuring each fits within the GPU's memory constraints. Other beneficial options would be improving computation efficiency, such as employing a lightweight backbone network (*e.g.*, TriVol [4]) or leveraging the autoencoding approach (*e.g.*, from [5]) for shape compression. These techniques could be incorporated into our generic pipeline to facilitate high-resolution shape completion.
>
> **Q6: For worse diversity, is it because diffusion models give more deterministic results than other methods?**
>
> A6: It's not about the diffusion model, as RePaint-3D, with the best diversity, is also diffusion-based. This is because our design choice prioritizes completion accuracy over diversity. Yet, the accuracy-diversity trade-off can be easily adjusted in DiffComplete. By altering the feature aggregation level, our method can achieve the best diversity, as shown in Table 7 of the main paper.
> ****
> **Reference**
>
> [1] Zheng, et al. Fast Sampling of Diffusion Models via Operator Learning. In NeurIPS 2022 Workshop.
>
> [2] Lugmayr, et al. RePaint: Inpainting using Denoising Diffusion Probabilistic Models. In CVPR 2022.
>
> [3] Zhou, et al. 3D Shape Generation and Completion Through Point-Voxel Diffusion. In ICCV 2021.
>
> [4] Hu, et al. TriVol: Point Cloud Rendering via Triple Volumes. In CVPR 2023.
>
> [5] Li, et al. Diffusion-SDF: Text-to-Shape via Voxelized Diffusion. In CVPR 2023.

---

### Official Review · Reviewer_DbPU · 2023-07-06

**Soundness:** 3 good
**Presentation:** 3 good
**Contribution:** 2 fair
**Rating:** 5
**Confidence:** 4

**Summary:**

The paper tackles the problem of probabilistic shape completion using diffusion models, learning from range scans. The main contribution of the paper comes from proposing 2 novel techniques: hierarchical feature aggregation for strong conditioning and occupancy-aware fusion technique. The method is tested on ShapeNet dataset and achieves better completions compared to the baselines. Also, the proposed method exhibits robust generatlization for out of distribution inputs.


**Strengths:**

* I enjoyed the idea of hierarchical feature aggregation. To best of my knowledge, conditioning a diffusion model is hard, and the ablation study in Tab. 7 clearly shows that the multi-level conditioning acts as intended.

* Strong empirical results on completion on unseen categories. The results in Tab2, 3 shows that the method achieves strong completion results on unseen categories and it makes sense due to strong hierarchical conditioning.


**Weaknesses:**

* Similar works on 3D completion using diffusion models exists. As mentioned in the related sections, the difference compes from the representation used to represent the state in the diffusion model. For example, point-voxel uses point cloud, while this work uses TSDF (TUDF) representation. While I do not think that the novelty of the work degrades even there are other diffusion based 3D completion models, I want to hear in detail how the method differs from other diffusion based models. Please look at the question section.

* My main concern is that some of the important recent works are not mentioned and compared. For determnisitc completion methods, convocc completes the given partial point cloud using occupancy fields. For multimodal completion methods, point-voxel [2] (although mentioned in the related-section) and GCA[3] completes point cloud in point cloud/sparse voxel representation. ShapeFormer[4] and cGCA[5] completes the point cloud in implicit neural representations.
Although I realize that comparing all of these baselines requires alot of effort, these baselines should have been tested since the core contribution of the work stems from achieving SOTA results on an existing benchmark.

[1] Peng et al. Convolutional Occupancy Networks. ECCV, 2020

[2] Zhou et al. 3D Shape Generation and Completion Through Point-Voxel Diffusion. ICCV, 2021

[3] Zhang et al. Learning to generate 3d shapes with generative cellular automata. ICLR, 2021

[4] Yan et al. ShapeFormer: Transformer-based Shape Completion via Sparse Representation. CVPR, 2022

[5]  Zhang et al. Probabilistic implicit scene completion. ICLR, 2022



**Questions:**

* Regarding the related work with diffusion based models, the authors mentioned that “. Due to the absence of meaningful ground truths in these completion scenarios, they could also face completion accuracy challenges like the above generative approaches” in line 114. Does that mean that the diffusion methods require the ground truth while the proposed method does not? To the best of my knowledge, the diffcomplete model requires ground truth TUDF for training. I think that the assumption that we can acquire the ground truth TUDF means that we can acquire ground truth point cloud and implicit function (mostly UDF based) as well. If so, then other baseline methods mentioned in the weakness section should be compared.

**Limitations:**

Authors have addressed the limitations in the supplementary material.

---

> ### Author Rebuttal · Authors · 2023-08-08
>
> Dear Reviewer DbPU,
>
> Thanks for the valuable feedback. We are happy to address each specific comment below.
>
> **Q1: Explain more details about the differences from other diffusion-based models.**
>
> A1: DiffComplete is specifically designed for high-quality shape completion. As a result, it differs from recent 3D diffusion models in various design choices, as summarized in **Table R7**: (i) diffusion Space, (ii) training strategy, and (iii) condition injection mechanism. Below, we present the details about each of these aspects.
>
> **(i) Diffusion Space**: Different from many previous approaches, We perform the diffusion process on the original volumetric TUDF. Doing so helps to ensure a direct transition to final meshes via Marching Cubes. This choice is favored as it preserves the fidelity of shape information better than the latent codes and circumvents the need for post-processing required by point clouds. While radiance fields are suited for rendering tasks, they fall short when recovering the shape geometries.
>
> **(ii) Training Strategy**: DiffRF [1] and Diffusion-SDF [2] both perform a masked diffusion process to incorporate conditions only at the inference stage. They do not include any conditional inputs at the training stage, thus lacking paired incomplete-to-complete GT supervision to accurately learn the completion rules. In contrast, Diffusion-SDF [3], PVD [4], and our method utilize these scan pairs in training to enhance the completion accuracy. However, we design different mechanisms to inject the conditions, as we discuss next.
>
> **(iii) Condition Injection**: Utilizing a control branch for hierarchical condition injection is a key design of DiffComplete, which enables both accurate completion and effective controls. This strategy yields superior performance over both the cross-attention (see Table 8 of the main paper) and the PVD's injection mechanism (see **Tables R1**, **R2**, and **R3** in *the response to all reviewers*, located at the top of this webpage).
>
> In the revision, we'll add these discussions into the supplement to clarify DiffComplete's unique attributes.
>
> **Table R7:** Comparisons with different 3D diffusion models. Note that [2] and [3] are different works.
> |**Method**|**Diffusion Space**|**Conditional Training**|**Condition Injection**|
> |-|-|-|-|
> |DiffRF [1]|radiance field|×|masked diffusion|
> |Diffusion-SDF [2]|latent patch|×|masked diffusion|
> |Diffusion-SDF [3]|latent vector|✓|cross-attention|
> |PVD [4]|point cloud|✓|single branch|
> |DiffComplete|volumetric TUDF|✓|two-branch hierarchical aggregation|
>
>
> **Q2: Confusions in Line 114 - "Due to the absence of meaningful ground truths in these completion scenarios, they could also face completion accuracy challenges like the above generative approaches".**
>
> A2: Similar to Diffusion-SDF [3] and PVD [4], our method requires incomplete-to-complete ground-truth shape pairs during training, as indicated in **Table R7**. For line 114, we intended to illustrate that certain 3D diffusion models (DiffRF [1] and Diffusion-SDF [2]) do not use these pairs for conditional training, which limits their completion accuracy. To avoid any confusion, the related sentences (lines 112-117) will be revised as follows:
>
> "For conditional shape completion, both DiffRF and Diffusion-SDF adopt a masked diffusion strategy to fill in large missing regions cropped out by 3D boxes. However, their training processes do not leverage a paired incomplete-to-complete ground truth, which may prevent them from accurately learning the completion rules. Contrarily, our method explicitly uses the scan pairs for conditional training."
>
> **Q3: Comparisons with the suggested baselines.**
>
> A3: As suggested, we have extensively compared DiffComplete with ConvONet, ShapeFormer, cGCA, and diffusion-based PVD. The results are illustrated in **Tables R1**, **R2**, and **R3** within *the response to all reviewers* with detailed discussions, located at the top of this webpage. Notably, DiffComplete exhibits obvious advantages over these baselines across various benchmarks and evaluation metrics. As GCA and cGCA share a similar generation paradigm, we only compared DiffComplete with cGCA, which has superior performance as demonstrated in its paper.
>
> ****
> **References**
>
> [1] Muller, et al. DiffRF: Rendering-Guided 3D Radiance Field Diffusion. In CVPR 2023.
>
> [2] Li, et al. Diffusion-SDF: Text-to-Shape via Voxelized Diffusion. In CVPR 2023.
>
> [3] Chou, et al. Diffusion-SDF: Conditional Generative Modeling of Signed Distance Functions. In ICCV 2023.
>
> [4] Zhou, et al. 3D Shape Generation and Completion Through Point-Voxel Diffusion. In ICCV 2021.

---

> > ### Comment · Reviewer_DbPU · 2023-08-16
> > **Revision**
> >
> > I indeed thank the authors for the clarification and all the experiments conducted. I feel that the paper is much stronger for it. It would be nice if the authors could visualize the qualitative results for the newly added baselines, but I think that can be done in the camera-ready version. I am convinced that the method works well compared to other baselines and I will raise the score.

---

> > > ### Author Response · Authors · 2023-08-18
> > > **Author Response to Reviewer DbPU**
> > >
> > > Dear Reviewer DbPU,
> > >
> > > Thank you for your positive acknowledgment and updated score. We've conducted visual comparisons with the newly-suggested baselines, including ConvONet, ShapeFormer, and PVD. Our method consistently demonstrates superior qualitative results for both known and unseen object categories.
> > >
> > > Due to NeurIPS 2023's guidelines that prohibit external links in the rebuttal box, we cannot show the visualizations here. Instead, we've shared an anonymized Dropbox link in the "Official Comment" section at the top of the review page for the AC's reference. We are happy to incorporate these visual comparisons into Figures 3 and 4 of the camera-ready paper. Our code will also be available to facilitate reproducibility.

---

### Official Review · Reviewer_KuxT · 2023-07-06

**Soundness:** 4 excellent
**Presentation:** 4 excellent
**Contribution:** 3 good
**Rating:** 8
**Confidence:** 3

**Summary:**

The paper introduces a diffusion-based approach, DiffComplete, to generate complete shapes conditioned on partial 3D range scans. The condition is represented as volumetric TSDF (truncated signed distance function), while the complete shape is represented as volumetric TUDF (truncated signed distance function). Inspired by 2D ControlNet, the authors devise a hierarchical feature aggregation mechanism to inject conditional features in a spatially-consistent manner. Besides, they propose a fine-tuning strategy to adapt the model trained on a single condition to multiple conditions. The trade-off between multi-modality and high fidelity can be controlled through the network level for feature aggregation between conditions and denoised complete shapes. The results on 3D-EPN and PatchComplete show the superiority of DiffComplete. The authors also demonstrate its zero-shot ability.

**Strengths:**

- The paper is clearly written and easy to follow.
- The ControlNet-style design to inject features from the condition is reasonable.
- Fine-tuning the network trained with a single incomplete shape for multiple incomplete shapes is a good strategy.
- The generalizability looks good according to Fig. 4.

**Weaknesses:**

It is unclear why some baselines (especially point-cloud-based methods) are missing. For example, "3D Shape Generation and Completion Through Point-Voxel Diffusion" and "SnowflakeNet: Point Cloud Completion by Snowflake Point Deconvolution with Skip-Transformer".

**Questions:**

1. Is there a typo in Figure 7? Is it an MMD curve?
2. For the ablation study on multiple conditions, can the authors first fuse the scans and compute TSDF, then extract features instead of averaging features extracted from individual scans?

**Limitations:**

The authors have adequately addressed the limitations (e.g., failure cases) in the appendix.

---

> ### Author Rebuttal · Authors · 2023-08-08
>
> Dear Reviewer KuxT,
>
> Thanks for the encouraging feedback. We are grateful to see your recognition of our methodology, model performance, and manuscript writing. We address the specific comments below.
>
> **Q1: Compare with the suggested baselines (especially point-cloud-based methods), e.g., PVD [1] and SnowflakeNet [2].**
>
> A1: In our rebuttal (see above), we've expanded comparisons with several suggested surface completion baselines, including an adapted version of PVD. Please refer to **Tables R1**, **R2**, and **R3** within *the response to all reviewers* for details, located at the top of this webpage. These additional comparisons further show the superior capability of our method and will be included in the final revision.
>
> Regarding methods on point cloud completion, we compare our DiffComplete with the leading approach SnowflakeNet (TPAMI 2022) as per your suggestion. To make the comparisons fair, we converted SnowflakeNet's outputs to meshes using the reconstruction technique from ConvONet [3]. As shown in **Table R5**, our method delivers much better completion accuracy. We'll include this table in the supplement of the revision.
>
> **Table R5:** Comparisons of completion accuracy on the 3D-EPN benchmark, evaluated by average $l_1$-error ($\downarrow$) across eight object classes.
> | **Method**|**Output**|**Avg. $l_1$-err. ($\downarrow$)**|
> |-|-|:-:|
> |SnowflakeNet [2]|point cloud|0.189|
> |DiffComplete (Ours)|volumetric TUDF|**0.053**|
>
> **Q2: Is there a typo in Figure 7? Is it an MMD curve?**
>
> A2: Figure 7 is a TMD curve to reveal the varying shape diversity. Yet, a typo in line 301 identifies it as an MMD curve. We have already corrected the issue and will do a careful wording pass.
>
> **Q3: For multiple conditional inputs, can the authors first fuse the scans and compute TSDF, then extract features instead of averaging features extracted from individual scans?**
>
> A3: We compare our strategy with this option in Section B.2 of the supplement. As indicated by **Table R6** (copy of Table 10 in the supplement), fusing the original scans might be vulnerable to registration errors, which can disrupt the final results. Instead, fusing scan features in hierarchical feature spaces shows greater resilience to simple noise at the TSDF level. For a more clear illustration of our design, we will move this ablation study to the main paper.
>
> **Table R6:** Choice of fusion space for multiple partial shapes. Directly fusing them in the original TSDF space significantly impairs the completion quality.
> |**Fusion Space**|**$l_1$-err. $\downarrow$**|**CD $\downarrow$**|**IoU $\uparrow$**|
> |-|:-:|:-:|:-:|
> |TSDF|0.12|4.78|61.0|
> |feature|**0.05**|**3.97**|**68.3**|
> ****
> **References**
>
> [1] Zhou, et al. 3D Shape Generation and Completion Through Point-Voxel Diffusion. In ICCV 2021.
>
> [2] Xiang, et al. Snowflake Point Deconvolution for Point Cloud Completion and Generation with Skip-Transformer. TPAMI 2022.
>
> [3] Peng, et al. Convolutional Occupancy Networks. In ECCV 2020.

---

> > ### Comment · Reviewer_KuxT · 2023-08-13
> >
> > Thank the authors for the extra comparison with missing baselines. The rebuttal has resolved my concern. I would like to keep my rating.

---

> > > ### Author Response · Authors · 2023-08-14
> > > **Author Response**
> > >
> > > Dear Reviewer KuxT,
> > >
> > > We sincerely thank you for your feedback and support. Your suggestions greatly help us refine our work.

---

### Official Review · Reviewer_Ct9H · 2023-07-07

**Soundness:** 2 fair
**Presentation:** 2 fair
**Contribution:** 2 fair
**Rating:** 5
**Confidence:** 4

**Summary:**

The proposed method aims to tackle the TSDF shape completion problem using diffusion models. That is, given one or several incomplete TSDFs obtained from partial scans of a single object produced by range sensors, the method generates a TSDF of complete shape while trying to preserve the geometric details of the incomplete one. The method achieves this by using the same technique in ControlNet: adding a control branch to the vanilla UNet module in diffusion models.
Also, in order to better incorporate the partial scans from different views, the authors propose an "occupancy-aware fusion" module which performs a weighted feature fusion for the multi-view scans considering their geometric reliability.

**Strengths:**

In general, the paper is easy to read and is technically sound.
The following are specific points:

- Single probabilistic framework for all tasks. As more and more recent papers suggested, generative models are not only doing great in generation problems, but are also competitive in deterministic problems where usually a unique optimal solution is desired given the input. This method shows this point for the 3D completion problem: Even when the input shape has little to no ambiguity, the generated completion has better quality than previous deterministic approaches.

- Incorporate multi-view scans. This special consideration is useful in real-life application, where the scanners are constantly capturing new scans and the small errors in registration will make it non-trivial to fuse features from different views.

**Weaknesses:**

I have concerns on baseline & references, unsatisfactory contricution and writing. Hence I believe this paper does not reach the NeurIPS bar and would like to rate "Reject".
Here are detailed points:

- Missing baselines and references. Comparison with multiple very related works are omitted. ConvOccNet can perform shape completion very well in higher resolution (64 and higher, see the Fig. 4 in their supplementary). The model is very good at deterministic completion. For multimodal setting, two very related work, ShapeFormer and 3DILG, are not mentioned or discussed. And the former work is especially designed for multimodal completion.

- Contribution is not satisfactory. The proposed method seems to be just deploying ControlNet on the PatchComplete problem and benchmark. The used resolution (32) is also not impressive comparing to previous works like DiffusionSDF or 3DShape2VecSet. I would be skeptical that training diffusion models directly and keep the partial tsdf unchanged during sampling (as what PVD does for completion) can achieve better results.

- Many minor writing problems. L99: It is confusing to classify AutoSDF to be a type of autoencoder. It is an autoregressive approach.
L114: "Due to the absence of meaningful ground truths, ...". In the mentioned works, they all have meaningful ground truths shapes. It is unclear what is "meaningful" here.
L325: "Effects of feature aggregation manner". A better way is to write: "Effects of the manner in which features are aggregated" or "Effects of our feature aggregation mechanism"

**Questions:**

- Please show advantages over these baselines:
(1) Convolutional Occupancy Network. Train ConvOccNet to map partial TSDF grid to complete grid. Then compare.
(2) ShapeFormer. TSDF can be converted to point cloud. Train ShapeFormer to map point cloud to implicit TSDF fields.
(3) TSDF diffusion with PVD style of completion sampling. That is, in the sampling time, always replace the noisy to the actual TSDF value in the place of partial scanned cells. This way, only the "missing" regions are denoised and completed.

**Limitations:**

Authors include a comprehensive discussion on method limitations and potential societal impact in the supplementary material.

---

> ### Author Rebuttal · Authors · 2023-08-04
>
> Dear Reviewer Ct9H,
>
> In the following, we address all comments in the review. Our additional results validate that our method achieves a clear improvement over the newly-suggested baselines. We are happy to include these experiments in the final revision of the paper.
>
> **Q1: Add comparisons with the suggested baselines.**
>
> A1: Comprehensive comparisons with ConvONet, ShapeFormer, and PVD are presented in **Tables R1**, **R2**, and **R3** within *the response to all reviewers*, located at the top of this webpage. These results show the superior performance of our method. Regarding the suggested 3D-ILG [1], it mainly focuses on the representation for shape generation rather than the specific completion task. As 3D-ILG also supports shape completion with a conditional generation paradigm, we compare it with our method with aligned data and experimental setups. As presented in **Table R4**, our method achieves much more accurate results compared to 3D-ILG. We'll include **Table R4** into our supplement in the revision.
>
> **Table R4:** Comparisons of completion accuracy on the 3D-EPN benchmark, evaluated by average $l_1$-error ($\downarrow$) across eight object classes.
> | **Method**|**Avg. $l_1$-err. ($\downarrow$)**|
> |-|:-:|
> |3D-ILG [1]|0.165|
> |DiffComplete (Ours)|**0.053**|
>
> **Q2: Is the proposed method a deployment of ControlNet on the PatchComplete benchmark?**
>
> A2: Although the paradigm of injecting conditional features takes inspiration from ControlNet, DiffComplete has critical distinctions to justify our contributions.
>
> First, DiffComplete differs from ControlNet in several key areas. **(i) Task:** ControlNet tackles the 2D text-to-image generation task, making it work on our 3D completion task is non-trivial. To this end, we design the appropriate volumetric representation and 3D networks. **(ii) Motivation:** The motivation of ControlNet is to finetune pretrained large diffusion models, while we aim to train a specific diffusion model. This leads to different training strategies. **(iii) Training Strategy:** ControlNet utilizes a "trainable copy" initialization, but our experiments found that training from scratch is the most effective way. Please see Table 9 in the supplement for details. **(iv) Design Details:** We discard "zero convolution", a critical component in ControlNet, as we do not require the pre-training process. We also directly embed the original 3D shape representation rather than operating in latent space.
>
> Second, DiffComplete offers new features and insights beyond ControlNet. **(i)** Our method further supports multiple inputs to improve the completion accuracy. **(ii)** We delve deeper into the feature injection mechanism, observing that altering the feature aggregation level finely controls the trade-off between completion accuracy and diversity. This finding can be leveraged to adjust the model's performance, as described in Section 4.5 of the main paper.
>
> **Q3: The used resolution ($32^3$) is not impressive.**
>
> A3: We are able to scale up the resolution through various available strategies. For instance, we can apply the gradient accumulation technique to break the large batch into smaller chunks, ensuring each fits within the GPU's memory constraints. Such a technique effectively facilitates the training of DiffComplete at higher resolutions. Other beneficial options would be improving computation efficiency, such as employing a lightweight backbone network (*e.g.*, TriVol [2]) or leveraging the autoencoding approach (*e.g.*, from [3]) for shape compression. These approaches could be incorporated into our generic pipeline to complete high-resolution shapes even on smaller GPUs.
>
> **Q4: Will training diffusion models with PVD paradigm achieve better results than DiffComplete?**
>
> A4: DiffComplete attains much better performance than PVD, as shown in **Tables R1**, **R2**, and **R3** within *the response to all reviewers* with detailed analysis. In addition, our method can flexibly support multiple conditional inputs, while this may not be feasible for the PVD pipeline.
>
> **Q5: Minor writing problems.**
>
> A5: Thanks for pointing out these wording issues and typos. We will carefully polish and revise the paper.
>
> **(i)** Though AutoSDF employs an AutoEncoder, it should be classified as an autoregressive model. We'll rectify this.
>
> **(ii)** Our goal is to illustrate that certain 3D diffusion models (Diffusion-SDF [3] and DiffRF [4]) do not employ incomplete-to-complete ground-truth pairs during training. The related sentences (lines 112-117) will be revised as follows:
>
> "For conditional shape completion, both DiffRF and Diffusion-SDF adopt a masked diffusion strategy to fill in missing regions cropped out by 3D boxes. However, their training processes do not leverage a paired incomplete-to-complete ground truth, which may prevent them from accurately learning the completion rules. Contrarily, our method explicitly uses the scan pairs for conditional training."
>
> **(iii)** Thanks for the suggestions. We've already fixed it.
> ****
> **References**
>
> [1] Zhang, et al. 3DILG: Irregular Latent Grids for 3D Generative Modeling. In NeurIPS 2022.
>
> [2] Hu, et al. TriVol: Point Cloud Rendering via Triple Volumes. In CVPR 2023.
>
> [3] Li, et al. Diffusion-SDF: Text-to-Shape via Voxelized Diffusion. In CVPR 2023.
>
> [4] Muller, et al. DiffRF: Rendering-Guided 3D Radiance Field Diffusion. In CVPR 2023.

---

> > ### Comment · Reviewer_Ct9H · 2023-08-16
> >
> > Thanks for the authors' clarification and the additional experiments. In general, the rebuttal addresses most of my concerns.
> > For Q2&A2, after seeing the authors explanation, I agree that adapt Control-Net for 3D completion problem require non-trivial effort.
> > For Q1&R1, I appreciate the authors' effort for the analysis and the extra quantitative comparisons. It would be better to show the comparisons visually, especially for ConvONet.
> >
> > In conclusion, I will raise my rating and am looking forward to seeing these updates incorporated in the revised paper.

---

> > > ### Author Response · Authors · 2023-08-18
> > > **Author Response to Reviewer Ct9H**
> > >
> > > Dear Reviewer Ct9H,
> > >
> > > Thank you for acknowledging our methodology and considering a score increase. In response to your suggestion, we have made additional visual comparisons with ConvONet, ShapeFormer, and PVD. The qualitative results showcase our method's superior completion quality for both known and ​unseen object categories.
> > >
> > > Due to the conference guidelines that prohibit external links in the rebuttal box, the visualizations cannot be displayed here. Yet, we've shared an anonymous link to our figures in the "Official Comment" section at the top of the review page for the AC's reference. We will incorporate these visual results into Figures 3 and 4 of the revised paper and release our code to facilitate reproducibility. We hope this reply addresses your remaining concerns and are looking forward to your final decision.

---

> > > > ### Author Response · Authors · 2023-08-21
> > > > **Following Response to Reviewer Ct9H**
> > > >
> > > > Dear Reviewer Ct9H,
> > > >
> > > > Thanks for your further comments and your consideration of our new updates.
> > > >
> > > > As the deadline for the author-reviewer discussion phase is approaching (Aug 21st at 1 pm EDT), we wish to inquire if our earlier post has further addressed your remaining concerns.
> > > >
> > > > In line with your feedback, we have made additional visual comparisons and have shared these results for the AC’s reference. We will incorporate all the updates mentioned in rebuttal into the revised paper. We will try to respond further before the discussion period ends, if there's anything more that we may do.
> > > >
> > > > We appreciate your time and consideration.
> > > >
> > > > Best regards,
> > > >
> > > > Submission 917 Authors

---

### Author Rebuttal · Authors · 2023-08-08

Dear all reviewers,

We sincerely thank your constructive comments and are grateful to see the recognition received:

- **Methodology:** Reviewer 9z1Q: "I like the adoption of ControlNet in the design"; Reviewer KuxT: "The ControlNet-style design is reasonable", "Fine-tuning the network for multiple incomplete shapes is a good strategy"; Reviewer DbPU: "I enjoyed the idea of hierarchical feature aggregation"; Reviewer Ct9H: "The paper is technically sound'.

- **Experimental Results:** Reviewer 9z1Q: "clearly outperforms other SOTAs", "shown their importance in the design through ablation studies"; Reviewer hw6P: "a new SOTA performance"; Reviewer KuxT: "The generalizability looks good"; Reviewer DBPU: "Strong empirical results on completion on unseen categories".

- **Practicality**: Reviewer 9z1Q: "Data is from the actual scan, making the method very useful and interesting"; Reviewer DbPU: "The stepwise addition of partial scans is interesting and makes completion tasks interactive and adaptable".

We first address the general concern of Reviewer Ct9H/KuxT/DbPU below. Then, we provide a detailed  response to each reviewer's comments and address all issues.

**Q: Performance comparisons with the suggested baselines. (To Reviewer Ct9H/KuxT/DbPU)**

A: The suggested baselines include both the deterministic approach (ConvONet [1]) and multimodal approaches (ShapeFormer [2], PVD [3], and cGCA [4]). We conducted experiments on both the 3D-EPN and PatchComplete benchmarks, showing the results in **Table R1** and **Table R2**, respectively, inside our rebuttal. Our DiffComplete surpasses all of these baselines by a significant margin with respect to completion accuracy and generalization ability. We provide a detailed discussion below.

 **(i)** ConvONet [1] extends the 3D-EPN structure (which is a baseline in our paper) by incorporating an implicit occupancy decoder. Both methods are deterministic by nature and hence cannot handle ambiguities of missing data. Thus, their completion accuracy and generalization ability are lower than our method.

 **(ii)** Our method differs from ShapeFormer [2] and cGCA [4] by the proposed generative model. The employed diffusion model offers superior sampling quality than the autoregressive model (ShapeFormer) and the Generative Cellular Automata model (cGCA). Compared with [2] and [4], which perform shape compression, we also preserve the fidelity of the original structures to enhance the completion details.

 **(iii)** PVD [3] is originally a point cloud diffusion model. Here, we adapt it to perform TSDF (TUDF) diffusion as suggested by Reviewer Ct9H. A limitation of PVD is that it retains noises of the partial input. Hence, when it is mixed with the generated missing part, noise severely affects the final completion quality. Instead, we design two branches to separately process the partial and complete shapes. By doing so, the model can effectively learn a diffusion process from noise to clean shapes.

For multimodal approaches, we further compare their multimodal capacity. **Table R3** shows that our method achieves significant improvements on both completion accuracy (MMD) and fidelity (UHD), while preserving a moderate (or even comparable) completion diversity (TMD). This result aligns with our design choice that prioritizes completion accuracy over diversity. We'd like to highlight that the trade off between accuracy and diversity can be easily adjusted in DiffComplete, as detailed in Section 4.5 of the main paper.

All experiments are conducted on the same data for a fair comparison. We'll include the additional comparison results from **Tables R1**, **R2**, and **R3** into Tables 1, 2, and 4 of the main paper, respectively, since they directly correlate. We will also release our code to facilitate reproducibility.

**Table R1:** Comparisons of completion accuracy on the 3D-EPN benchmark, evaluated by average $l_1$-error ($\downarrow$) across eight object classes.
| **Method**|**Paradigm**|**Avg. $l_1$-err. ($\downarrow$)**|
|-|-|:-:|
|ConvONet [1]|deterministic|0.220|
|ShapeFormer [2]|multi-modal|0.141|
|PVD [3]|multi-modal|0.114|
|cGCA [4]|multi-modal|0.185|
|DiffComplete (Ours)|multi-modal|**0.053**|

**Table R2:** Comparisons of generalization ability on the PatchComplete benchmark, evaluated by average CD ($\downarrow$) and IoU ($\uparrow$) across eight *unseen* object classes.
| **Method**|**Paradigm**|**Avg. CD ($\downarrow$)**|**Avg. IoU ($\uparrow$)**|
|-|-|:-:|:-:|
|ConvONet [1]|deterministic|5.26|60.1|
|ShapeFormer [2]|multi-modal|5.05|62.5|
|PVD [3]|multi-modal|4.94|62.8|
|cGCA [4]|multi-modal|5.09|61.7|
|DiffComplete (Ours)|multi-modal|**4.10**|**67.5**|

**Table R3:** Comparisons of multimodal capacity on Chair class of the 3D-EPN benchmark.
| **Method**|**MMD ($\downarrow$)**|**TMD ($\uparrow$)**|**UHD ($\downarrow$)**|
|-|:-:|:-:|:-:|
|ShapeFormer [2]|0.007|0.024|0.055|
|PVD [3]|0.007|**0.027**|0.042|
|cGCA [4]|0.006|0.024|0.047|
|DiffComplete (Ours)|**0.002**|0.025|**0.032**|

**References**

[1] Peng, et al. Convolutional Occupancy Networks. In ECCV 2020.

[2] Yan, et al. ShapeFormer: Transformer-based Shape Completion via Sparse Representation. In CVPR 2022.

[3] Zhou, et al. 3D Shape Generation and Completion Through Point-Voxel Diffusion. In ICCV 2021.

[4] Zhang, et al. Probabilistic implicit scene completion. In ICLR 2022.

---

### Decision · Program_Chairs · 2023-09-21

**Decision:**

Accept (poster)

**Comment:**

The paper proposed a framework for shape completion with a conditional 3D diffusion model utilizing a control-net style conditioning, with the improvement of using hierarchical feature aggregation, and occupancy aware fusion. Multi-modal results and generalization to unseen categories demonstrate the effectiveness of the proposed method. The rebuttal and following discussions successfully addressed the concerns of all the reviewers who all became positive after the rebuttal, thus, I follow the reviewers' suggestions on accepting this paper.

As promised during the rebuttal and discussion, please add the additional comparisons with baselines provided during the rebuttal to the main paper, motivation and clarification on the feature aggregation schemes and exact experimental settings, and the discussion of extending to part-based generations.